# General mechanisms of task engagement in the primate frontal cortex

Jan Grohn [1] ✉, Nima Khalighinejad [1], Caroline I Jahn[1,2],
Alessandro Bongioanni [1,3], Urs Schüffelgen[1], Jerome Sallet[1,4],
Matthew F. S. Rushworth [1] & Nils Kolling [4,5,6]

Staying engaged is necessary to maintain goal-directed behaviors. Despite this, engagement exhibits continuous, intrinsic fluctuations. Even in experimental settings, animals, unlike most humans, repeatedly and spontaneously move between periods of complete task engagement and disengagement. We, therefore, looked at behavior in male macaques (*macaca mulatta)* in four tasks while recording fMRI signals. We identified consistent autocorrelation in task disengagement. This made it possible to build models capturing task-independent engagement. We identified task general patterns of neural activity linked to impending sudden task disengagement in mid-cingulate gyrus. By contrast, activity centered in perigenual anterior cingulate cortex (pgACC) was associated with maintenance of performance across tasks. Importantly, we carefully controlled for task-specific factors such as the reward history and other motivational effects, such as response vigor, in our analyses. Moreover, we showed pgACC activity had a causal link to task engagement: transcranial ultrasound stimulation of pgACC changed task engagement patterns.

Everyone experiences fluctuations in how engaged they are with tasks that need doing throughout the day. While some of our motivation is clearly linked to specific tasks and incentives, we also find ourselves from time to time either demotivated or full of vigor regardless of the task at hand. Furthermore, while there might be extended periods of disengagement, there are also brief collapses in task engagement (for example, while checking one's phone). While we also experience fluctuating levels of task engagement, in some people, periods of disengagement are especially prominent; apathy—sustained periods of task disengagement—is a core, transdiagnostic feature of psychological and neurological illnesses[1,2].

Such fluctuations occur even though engagement must be sustained across extended periods of time for many goal-directed behaviors to succeed. Additionally, when performing a task, it is important

to stay engaged independently of the specifics of the task at hand. Important insights into related processes have been gained by investigating motivation changes occurring in response to specific external factors such as reward incentives or other feedback[3]. However, task engagement is also subject to intrinsic fluctuation and must be maintained despite adverse external factors. Likewise, sometimes engagement is lost despite the presence of incentives. It has been proposed that maintaining engagement requires cognitive resources that are depleted by effort and that can be restored by taking breaks[4].

Changes in response vigor[5] and speed[6–13] occur as motivation waxes and wanes. However, variation in response vigor and speed occurs only if a person decides to maintain task engagement. Therefore, deciding whether or not to engage in the task at all or to pause and disengage completely is a separate process from the one

[1]Wellcome Centre for Integrative Neuroimaging (WIN), Department of Experimental Psychology, University of Oxford, Oxford, UK. [2]Princeton Neuroscience Institute, Princeton University, Princeton, NJ 08540, USA. [3]Cognitive Neuroimaging Unit, CEA, INSERM, Université Paris-Saclay, NeuroSpin Center, 91191 Gif/Yvette, France. [4]Université Lyon 1, Inserm, Stem Cell and Brain Research Institute U1208, 18 Avenue Doyen Lepine, 69500 Bron, France. [5]Wellcome Centre for Integrative Neuroimaging (WIN), Department of Psychiatry, University of Oxford, Oxford, UK. [6]Centre Hospitalier Le Vinatier, Pôle EST, Bron, France. ✉e-mail: jan.grohn@psy.ox.ac.uk

determining response speed and vigor for any given response. Similarly, task engagement differs from attention lapses as indexed by individual erroneous responses that have also previously been studied in the context of motivation[14].

In the present study, we focus on general mechanisms of task engagement and disengagement across a series of four different tasks while recording brain activity using fMRI. In this way, we can identify neural activity changes in moments when an agent spontaneously and completely disengages from a task independently of the concurrent specific, external task demands. We used macaque monkeys to examine these issues for several reasons. The social and other demands of human neuroimaging experiments usually ensure that human participants exhibit continuous task execution; their performance scores may fluctuate, but human participants rarely give up and spontaneously stop altogether in the same manner that they do frequently when outside the laboratory. Macaques, however, while engaged for the majority of the experiments, repeatedly and reproducibly both disengage and re-engage for periods of time during daily testing in the laboratory, even when the tasks are relatively simple and are performed proficiently[15,16]. While this is generally a great nuisance for the researchers, for our study it is fortunate as it allowed us to construct and fit models to these disengagements and link them to their neural substrates. Using data from four diverse decision-making tasks allows us to find behavioral and neural signatures that are task-general (see the Supplementary Note for descriptions of the four tasks). Importantly, these disengagements are not part of the task design but occur spontaneously despite the reward incentives provided by the tasks. Moreover, by controlling for variation in extrinsic experimental factors, such as reward level, we can capture engagement and disengagement due to task-independent factors. Intrinsic motivation has previously been linked to satiation (for example, cumulative reward[17], or time spent on task, e.g., ref. [18]). By also controlling for these factors, we aim to capture the intrinsically fluctuating aspect of task engagement and disengagement that occurs regardless of task identity[19].

While task engagement is continually fluctuating during extended activity[20] disengagements are all or none events. For example, one might feel more or less motivated to do a chore throughout the day—which we refer to here as the level of general task engagement. In addition, there are periods of complete cessation and disengagement from the task. We examined neural activity related to both slow fluctuations in engagements and sudden disengagements. To do this, we used an approach that considers the distribution of task engagements and disengagements to estimate continual variation in a general state of task engagement. Such a state tracks the current level of engagement above and beyond the current trial. This allowed us to identify events when animals suddenly and 'surprisingly' disengage even though they are in an otherwise engaged state. By contrast, we can also identify 'expected' disengagements that occur when we estimate that the animal is in a state of low general engagement. This allowed us to examine the neural activity linked to general task engagement, expected task disengagements, and surprising task disengagements. We argue that such model-derived estimates capture aspects of task engagement not previously reported in the literature: By linking engagement both to trial and state activity and estimating its task-independent component as our model is based on unexplained residual variance, we are able to parse aspects of task engagement not previously studied. Importantly, we contrasted these model-derived estimates of engagement with other distinct aspects of motivation, such as changes in response vigor indexed by reaction time. This made it possible to dissociate signals leading to task engagement or disengagement from neural activity related to variation in motivation to execute a specific action quickly.

By using a whole-brain imaging technique such as fMRI, we can seek neural correlates of engagement throughout the brain during all four tasks. This is important as the neural circuits linked to task engagement/disengagement are not well defined. However, we note that areas of the anterior cingulate cortex (ACC) and adjacent medial frontal cortex have been linked to intrinsically motivated behaviors[21], mood fluctuation[20], and neural activity has been reported to change in some related situations[15,22], particularly when driven by endogenous factors such as satiety[23].

Our fMRI analysis identified one important area of activity change in perigenual ACC (pgACC) that was prominent across all four tasks. We therefore used neurostimulation data in which activity in this region was manipulated to test its causal importance for task engagement: Specifically, one of the datasets used in our analysis had stimulated pgACC using transcranial ultrasound stimulation (TUS), and thus allowed us to compare the effect of pgACC stimulation against other control regions. Not only did we examine the impact of TUS on pgACC and compare it to sham TUS, but in addition, we also examined the impact of TUS on the basal forebrain (BF). BF TUS leads to changes in motivation-related influences on action timing[13] and so it provides an especially strong comparison with pgACC TUS. In addition, we examined the impact of TUS of an additional control region in the parietal operculum (POp)—a region in which task-related and task-initiation-related activity had not been observed—to control for general cortical stimulation effects.

## Results

We combined data from four different reward-based decision-making tasks[13,24–27]. The tasks covered a range of different paradigms: simple stimulus-response mapping, incentivized exploration/exploitation, incentivized delayed responses, and novel value inference (see the Supplementary Note for descriptions of the four tasks). In each case, the animals occasionally disengaged from the task and stopped responding before re-engaging after some time. For the purpose of our analysis, we define disengagements as responding after 3 s or later, or not responding at all during a trial, i.e. the trial "timed out" before a response was made. However, for one of our tasks that incentivized late responses[13,27], we only counted trials as disengaged where the animal did not respond at all (see Fig. S1 for details for all tasks). We binarized trials into ones where the animals are engaged or disengaged (Fig. 1A). This definition of disengagements conceptualizes behavior as all or no events, which we can contrast with a continuous measure of response vigor i.e., when the animals remain on task but respond more or less rapidly (see below). While other definitions of disengagement might be possible (e.g., by looking at decision errors), those would have not been applicable in our tasks due to the large variations in difficulty across task and because errors can occur during learning as well as when there is disengagement tasks. By applying our response time-based definition, we can consistently classify disengagements across a range of diverse tasks and capture the intrinsic, task-independent nature of these events. Our threshold of 3 s was chosen to ensure that on trials that were classified as disengagements, the animals made the decision to disengage rather than responding sluggishly while still being on-task. Overall, we started with 17 datasets in 13 animals but excluded six datasets from five animals that disengaged in <5% of trials on average across sessions (Fig. S1). Two animals that provided in one task[24] also provided data in two other tasks[13,27], which left eleven datasets from nine unique animals (see Fig. S1 for details).

Our aim was to use disengagements to construct variables that, on a trial-by-trial basis, capture different aspects of task engagement that are independent of the specific task identity. We then used these variables in an fMRI analysis to identify their neural correlates.

To contrast task engagement and disengagement with variation in motivation related to response vigor and speed, we repeated the same analysis using response times (RTs). For this control analysis we only used data on engaged trials (we did not analyze the trials classified as disengagements in which, by definition, no response or delayed response is made; see Fig. S1). For this analysis, we used data from 13

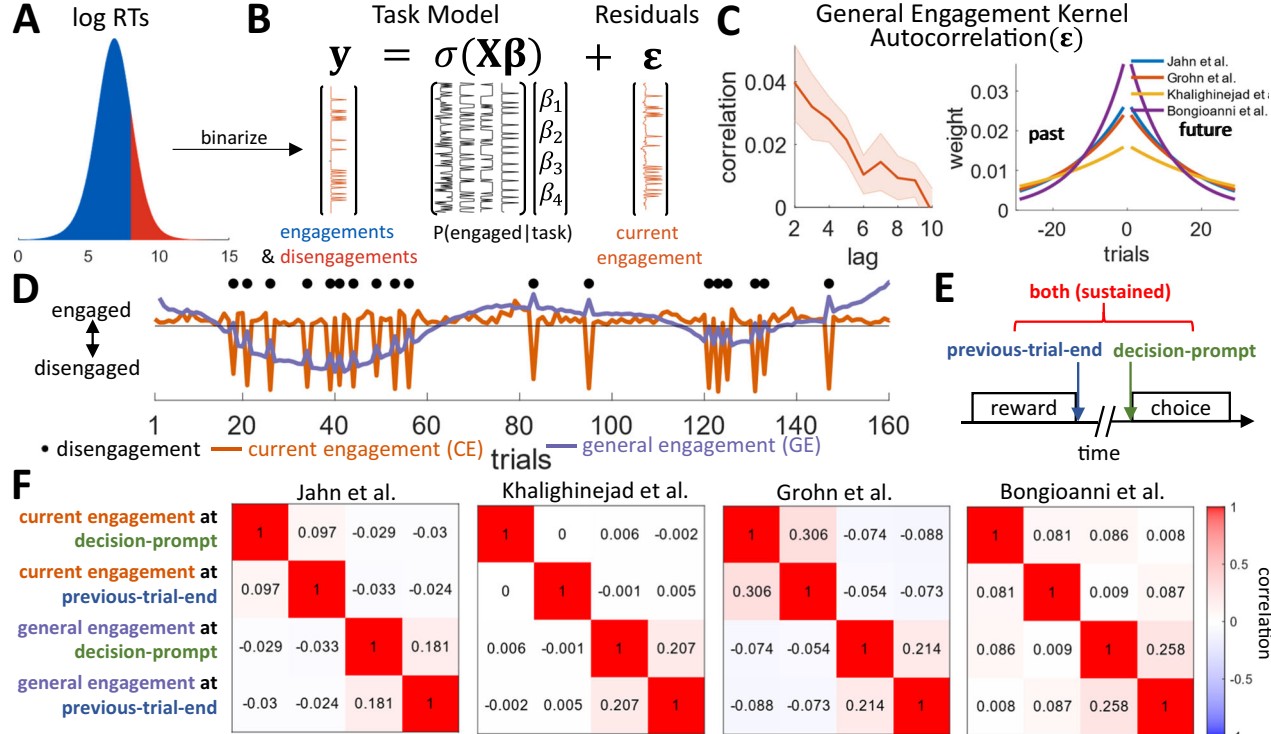

**Fig. 1 | Behavioral results and fMRI design. A** We binarized animal's RTs into trials in which they were engaged or disengaged. On disengaged trials, the animals took longer than 3 s to respond, or did not respond at all (i.e., the trial timed out). **B** To control for the influence on motivation exerted by extrinsic task event-related factors, we constructed separate logistic regression models for each of our four tasks. Each model contained regressors coding for other motivational effects and task-specific regressors (see *Methods* for details). By regressing the effects these variables have on engagement, we were left with the residuals. These residuals contain fluctuations in task engagement that are intrinsic as opposed to those that are due to extrinsic factors related to task structure and task events. We refer to this index as the intrinsic level of current engagement (CE). **C** (Left) We find a persistent autocorrelation of the residual fluctuations suggesting that intrinsic CE–engagements and disengagements–are temporally clustered. Shaded error represents the standard error of the mean across data sets. (Right) By fitting exponential kernels to the index of the intrinsic CE for each of the four tasks, we can also capture this autocorrelation. **D** The same kernels can then be used to smooth the estimate of the intrinsic CE (orange line, shown for an example session) on each trial in each task. As a result, an estimate is obtained of the slowly fluctuating general engagement (GE) of an animal that can be made available for each trial (purple line, shown for an example session). **E** To capture effects of task engagement in a similar manner in our neural analyses of all four tasks, we time-locked to two events in each trial that all our four tasks have in common: the end of the reward delivery in the previous trial, and the onset of the decision-prompt in the current trial. **F** Even after their hemodynamic convolution with the relatively fast hemodynamic response function observed in macaques[29,30], there is limited correlation between these regressors in all four tasks. Source data are provided as a Source Data file.

(unique) animals because we now had sufficient data from more animals to include in the analysis. However, we avoided considering data from one of the previous tasks[13,27] because the animals performing it were sometimes incentivized to respond late as part of the task design and thus RTs do not provide the simple measure of vigor in the same way as in other tasks.

## Behavioral results

For each task, we constructed separate regression models that accounted for the extrinsic variables that could be measured in each experiment by the investigators. These models included regressors such as the task stimuli encountered, the responses made, the rewards animals received, and the trial number (see *Methods* for the specific models for each task). Using these models, we can account for variance in task-engagement and disengagement that is due to extrinsic factors. These regressors are, of course, the ones that are usually the focus of any analysis of a neural data set. However, by regressing out the variance due to all extrinsic factors (i.e., taking the residual error of the regression models) we are left with the components of task-engagement and disengagement that are due to what is normally considered residual fluctuations in behavior that typically receive little investigative attention (Fig. 1B). However, these residuals also capture task engagement and disengagement that is dependent on intrinsic variation. As such, they capture the intrinsic level of *current*

*engagement* (CE; the distributions of CE for each task are shown in supplementary Fig. S2). Using the same analysis approach across tasks is essential for generalizability but also means we had to find a definition of disengagement that works across studies. Thus, while there might be some adjustment in the behavioral definition that could be made if we had only analyzed a single task, we employed an approach with the merit of general applicability; while we might have failed to detect task-specific motivational factors, the approach achieves the aim of identifying neural processes common to many situations.

If engagement is indeed drifting across trials, then we should be able to observe clustering in the residuals. To this end, we examined its autocorrelation. If engagement and disengagement were solely determined by extrinsic task features, then the residuals would not be autocorrelated over trials. However, in our data, we did indeed find persistent autocorrelation in the residuals, thus providing evidence for CE (Fig. 1C left; significant for lags <10 at $p < 0.05$ with Bonferroni correction; we exclude lag = 1 because in some tasks repeated disengagements were impossible, as the experiment waited for the animal to re-engage before continuing). In other words, periods of engagement and disengagement are temporally clustered. We confirmed that this is not an artefact of the regression models we used by randomly shuffling which trials are classified as engaged or disengaged and repeating this analysis 1000 times. Here, we did not find any autocorrelation of the residual over trials.

We can use the autocorrelation of CE to estimate the level of task engagement for each animal on each trial. We refer to this variable as *general engagement* (GE). While CE corresponds to the residual fluctuations in Fig. 1B, GE is a more general and slowly varying estimate of task engagement that is a weighted average of CE on the current but also on surrounding trials: if the animal disengages on previous/future trials, we can assume it is also, to some degree, in a disengaged state currently. Conversely, if it is engaged on these trials, we can assume it is also, to some degree, in an engaged state currently. To this end, we fit exponential kernels to the residual fluctuations (Fig. 1C right shows the fitted kernel for each of the four tasks). These kernels capture the extent to which task engagement on a trial, as indexed by the residual fluctuations, is related to task engagement on preceding and following trials. Smoothing the residual fluctuations (CE; orange line in Fig. 1D; shown after normalizing) by these kernels allows us to obtain an estimate of a continuously varying GE (blue line in Fig. 1D; shown after normalizing) on each trial. We construct GE this way to obtain an interpretable regressor we can use in our fMRI analyses. While CE and the disengage choices are closely (inversely) related, CE values are impacted by the degree of predictability of a specific disengagement choice (black dots in Fig. 1D vs orange line in Fig. 1D), and are thus also useful interpretable regressor for our fMRI analyses.

We can also combine the estimates of CE and GE to obtain two derived quantities that are used in first stages of the neural analysis as contrasts. First, we can average the current CE index with the continuously varying GE index to obtain an estimate of a third variable we refer to as *overall engagement* (OE). OE provides an overarching estimate of engagement on any trial as it uses both the engagement on the current trial (as given by CE) and of the surrounding trials (as given by GE) to index engagement, and so it is a useful starting point for neural analyses; as explained in more detail below, we can first identify areas in which activity is related to OE and then we can examine whether the activity tracks CE, the more slowly varying GE, or both quantities. Thus, CE and GE can also be thought of as the separated trial and state components of an overarching model that indexes OE. Second, we can subtract the model-derived estimate of GE from the CE level to identify *engagement shifts* (ES) when an animal's task engagement suddenly collapses and there is abrupt disengagement; the animal may be disengaged on the current trial even though the events that normally surround a disengagement were not observed. This allows us to examine CE when it is unexpected given the current level of GE; i.e. it allows us to identify trials with low engagement in an otherwise highly engaged state. Importantly, for the purpose of our neural analysis, both ES and OE can be constructed by subtracting/adding CE and GE on the contrast-level within a single general linear model.

We repeated an analogous, control analysis of RTs—an index of motivational change in relation to response vigor as opposed to task engagement. However, this analysis was performed on engaged trials only; responses were only made, and RTs were only measurable on engage trials (Fig. S3A–C). We again find that, after having regressed out the variance in RTs due to task-manipulations, the error in RT estimates is autocorrelated over trials (significant for lags <8 at $p < 0.05$ with Bonferroni correction). We refer to these residual fluctuations as *trial vigor*. By fitting exponential kernels to trial vigor, we again obtain estimates of a general *state vigor* on each trial. The GE and general *state vigor* estimates are analogous state-related variables but they are only weakly correlated (Fig. S3D) and thus reflect different potential motivational processes. Just as for ES and OE, we can also consider individual *trial vigor* (as explained above) and slow fluctuations in trial vigor—*state vigor*—to obtain analogous contrasts relating to response speed as opposed to task engagement to use in our neural analysis. Once again these vigor-related variables were uncorrelated with our key task engagement/disengagement related variables of interest.

## fMRI results

As in the behavioral analyses, we constructed a separate neural regression model for each task that captured all aspects of the extrinsic task variables (see *Methods* for the specific models). In addition to these task-specific models, we also included regressors that captured the task engagement factors that we identified in our behavioral analysis (Fig. 1C), and regressors accounting for body and limb motion during task-performance and low-quality volumes (see *Methods* for details). Because the neural activity we are interested in is related to overarching engagement that is not necessarily associated with any one event that occurred during the task, we time-locked our regressors to two separate points within each trial that all four tasks had in common: (1) we time-locked to the decision-prompt on each trial when monkeys were asked to make a choice, and (2) we time-locked to the end of the outcome-period of the previous trial when animals either received a reward or no reward for their previous choice[28]. This ensured we had a measure of activity when task-specific performance and learning in a trial had been concluded and potential preparatory activity for the coming trial was beginning while also ensuring that the measurement was taken in the same way across all tasks; the same two time points could be defined in an identical manner for all four tasks. Moreover, the previous-trial-end and the following decision-prompt are far enough apart in time to ensure that regressors time-locked to each event are relatively uncorrelated even after convolution with the macaque's fast hemodynamic response function[29,30] (Fig. 1F). We hypothesized that general task engagement-related activity—our signals of interest—should be found at both time points. In our analysis we, therefore, included regressors for both CE and GE at both time-points, and use contrasts to also estimate OE and ES. Moreover, we also included our analogous control estimates of the *trial vigor* level and the *state vigor* at both of the same time-points (Fig. S3). Importantly, as we can only estimate *trial vigor* and *state vigor* on engaged trials, these regressors are zeroed out on disengaged trials.

We combined the results of these session-level regressions separately for each data set per animal using fixed effects. In a final step, we combined the data from all data sets on a third level using random effects. This allows us to examine the neural correlates of task-independent engagement across tasks and animals. To examine the effects of engagement/disengagement we used eleven data sets from nine animals across four tasks while controlling for response vigor. Statistical significance was determined using a standard cluster-based thresholding criteria of $z > 2.3$ and $p < 0.05$[31]. Significant clusters for our contrasts of interest are shown as white outlines in Fig. 2. Additionally, we also show the non-cluster-corrected z-statistics at a lower threshold of $z > 1.5$ in Fig. 2 to give a more complete picture of the results. Moreover, in the supplementary analyses we report analyses for vigor-related effects using a larger sample of data from thirteen animals across three tasks, as discussed above.

When we examined neural activity related to CE (Fig. 2A), we saw a large overlap between activity at previous-trial-end (Fig. 2A left) and decision-prompt (Fig. 2A middle), with activity at decision-prompt being slightly more lateral. Combining these estimates allowed us to identify regions that show activity both at previous-trial-end and decision-prompt (Fig. 2A right), which suggests that it is sustained throughout this task period and not linked to any particular task event (Fig. 1E). While there was widespread activity in the brain, within frontal cortex, pgACC (area 32), ventromedial PFC (areas 25 and 14), and the larger orbitofrontal network (areas 12 and 13) were particularly active. For a full table of cluster locations and descriptions see Table S1.

Similarly, when we examined neural activity related to GE (Fig. 2B), we again saw a large overlap between activity at previous-trial-end (Fig. 2B left) and decision-prompt (Fig. 2B middle). Combining both time-points again yielded regions that show sustained activity (Fig. 2B right). While the activity again included pgACC (area 32) prominently, there was somewhat less ventromedial PFC and OFC

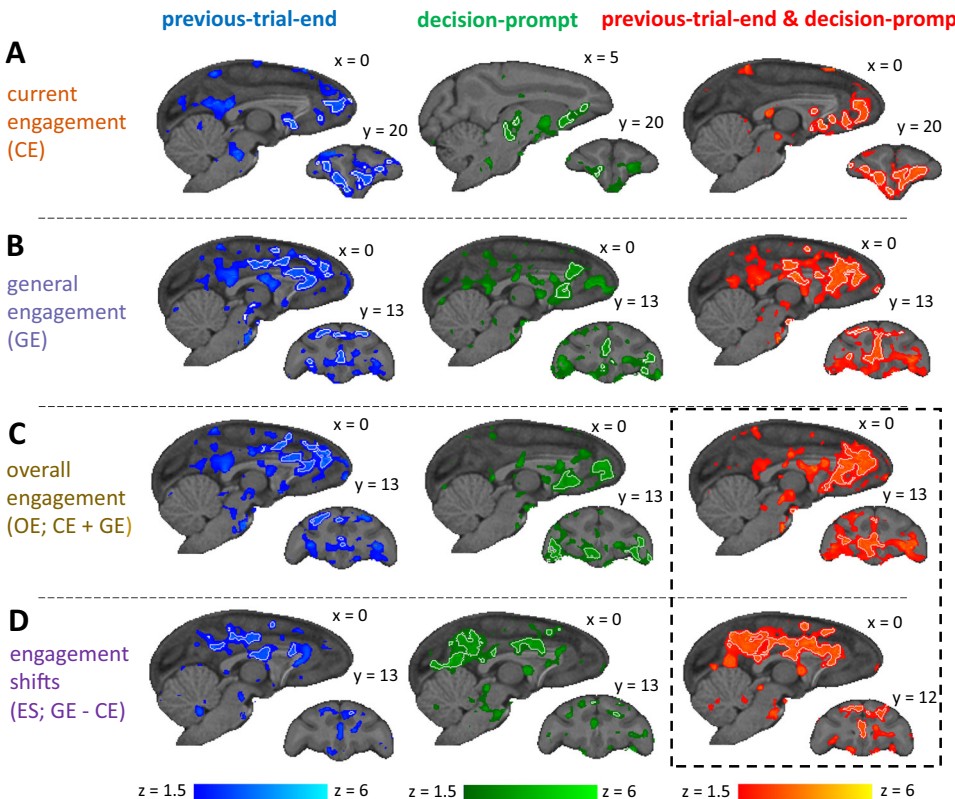

**Fig. 2 | Neural activity associated with engagement and disengagement.** Whole-brain activity is shown for different contrasts (top to bottom), time-locked to different events (left to right). Activity with z > 1.5 is shown superimposed, with white outlines indicating significant clusters at z > 2.3. **A** For CE, we observed activity in regions spanning pgACC (area 32), sgACC (area 25), and OFC (areas 12 and 13), both at previous-trial-end and decision-prompt and when looking at both time-points combined. **B** For GE, we observe activity throughout anterior and mid-cingulate gyrus (including pgACC and supracallosal gACC), and frontopolar cortex. **C** For OE,

we observed activation most prominently in pgACC but extending into adjacent sgACC and dACC, and also OFC areas 13 and 47/12o when animals are engaging with the current trial while also being in an overall engaged state. **D** For ES, we observed activity in the supracallosal cingulate cortex (including supracallosal gACC) when animals, surprisingly, disengaged from the trial despite otherwise being in an engaged state. Images in the figure were created by the authors using the McLaren template as implemented in the MrCat toolbox. Copyright: 2008-2011, Donald McLaren https://www.gnu.org/licenses/gpl-3.0.en.html.

activity and instead more activity in anterior supracallosal ACC gyrus (gACC; area 24) as well anterior dorsal ACC sulcus. Moreover, we found a significant cluster in frontopolar cortex (area 10o). For a full table of cluster locations and descriptions see Table S2.

To identify regions that were active when the animals had a high overall task engagement level, we combined our estimates of CE and GE into OE (Fig. 2C). At the end of the previous trial, activity was prominent in pgACC (area 32) and extended caudally into gACC (area 24) and into dorsal ACC sulcus (rostral cingulate zone) (Fig. 2C left). At decision-prompt, activity was again seen in pgACC (area 32), but otherwise more orbitofrontal (area 47/12o) (Fig. 2C middle). When combining activity at previous-trial-end and decision-prompt to find areas that were active throughout the whole task-period and across CE and GE, we observed a prominent and extensive area centered on pgACC (area 32), but extending into adjacent dorsal ACC sulcus (dACC; note that this area is sometimes referred to as mid-cingulate cortex or rostral cingulate zone) and subgenual ACC (sgACC; area 25) and also, albeit to a more limited extent in orbitofrontal cortex (OFC) in area 13 and the sub-region bordering ventrolateral prefrontal cortex −47/12o −, and striatum (Fig. 2C right). For a full table of cluster locations and descriptions see Table S3.

We also looked for effects of ES, i.e., the difference between GE and CE (Fig. 2D). Such activity was prominent when animals disengaged on the current trial while otherwise having been in an engaged state and likely to soon return again to an engaged state. In other words, the analysis identifies 'surprising' disengagements, where the disengagement is not preceded or followed by other disengagements;

or conversely engagement in a disengaged state. It thus identifies trials where our GE and CE indexes are opposed. Again, similar regions were active when time-locking to previous-trial-end (Fig. 2D left) and decision-prompt (Fig. 2D middle). When we time-locked to both previous-trial-end and decision-prompt, activity was prominent throughout mid supracallosal cingulate gyrus (area 24) (Fig. 2D right) extending into poster cingulate cortex and the precunous. For a full table of cluster locations and descriptions see Table S4.

Overall, while we saw some small differences between the focus of activation between previous trial end and decision prompt, none of the frontal effects were statistically different in a comparison between the two. All statistically significant differences we found were in more posterior parts of the brain, suggesting that the frontal circuit activity carrying general task engagement information is particularly sustained.

To further examine the factors driving engagement on the whole-brain level, we focused on activity that was present both at previous-trial-end and decision prompt (Fig. 2 right column) as this activity is most likely due to sustained task engagement. There we focused on OE-related and ES-related activity (Fig. 2 dotted lines) and extracted the BOLD time course from regions of interest (ROIs) we placed in grey matter within the areas of functional activity. Specifically, we defined the ROIs as the overlap between functional activity and anatomically defined regions (pgACC, OFC, striatum, and gACC)[32], and looked separately at the effects of CE and GE in the timecourse.

We observed that activity related to CE and GE appears similar in pgACC, OFC and the striatum (Fig. 3A–C middle rows). Activity related

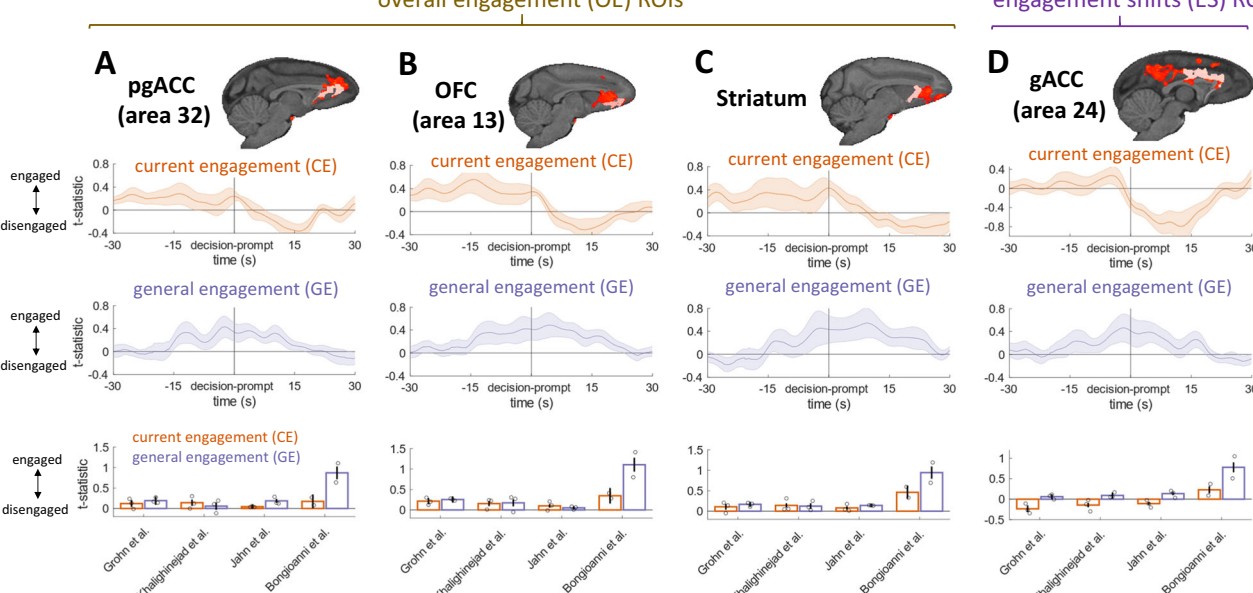

**Fig. 3 | CE and GE timecourses in ROIs.** We extracted timecourses from ROIs placed in anatomically defined regions within our significant OE and ES clusters for activity both at the previous trial end and decision-prompt. Significant clusters are shown in red with ROIs shown in light red (top). We then visualized the CE and GE timecourses in these regions time-locked to decision-prompt (middle rows). Shaded error bars represent standard errors of the mean across 11 animal datasets. We also extracted the t-statistics associated with CE and GE from our whole-brain analysis in the same ROIs to visualize the effects for each task separately (bottom row). Bars represent task means, and dots represent individual animal means, with error bars representing the standard error of the mean across sessions (**A**–**C**). Extracted CE timecourses from pgACC, OFC, and striatum show sustained activity

before and during the trial. By contrast, GE timecourses show sustained activity both before and after the trial. Effects are consistent across all four tasks (bottom). **D** Extracted CE timecourses from supracallosal gACC exhibit decreases during and after the current trial when animals disengaged, while GE timecourses are sustained increases beginning many seconds before and continuing many seconds after the current trial (i.e., engaged). Effects are consistent across three of the four tasks, with CE having the opposite (positive) sign in the fourth task (bottom). Source data are provided as a Source Data file. Images in the figure were created by the authors using the McLaren template as implemented in the MrCat toolbox. Copyright: 2008-2011, Donald McLaren https://www.gnu.org/licenses/gpl-3.0.en.html.

to GE extended over a window of ~30 s−15 s before and 15 s after the current trial. In contrast, activity related to current CE level was prominent before and on the trial itself. However, activity tracking both the more phasic CE level and the more tonic GE was observed across all areas in which OE effects were observed, namely pgACC (area 32), OFC (area 13), and striatum (Fig. 3A−C). Finally, to confirm that OE effects in each region were not driven by activity recorded just in one task, we extracted the t-statistics in these ROIs from the whole-brain analysis and examined them for differences by task (Fig. 3A−C bottom rows). Effects in the same direction were present in all four tasks and ROIs, although they were especially prominent in a task that required animals to make novel decisions[24].

Extracting the timecourse from the gACC ROI placed within the significant ES cluster (Fig. 3D) demonstrated that there was both a decrease in activity that was related to CE−an effect that began shortly before trial onset but which was then sustained for some time afterwards−and an increase in activity related to GE (Fig. 3D middle). To confirm that the effect was not driven by any one particular task, we extracted the t-statistics in the ROIs identified by the whole-brain analysis and examined them by task. We found broadly similar effects in three tasks although the current CE effect was different in the fourth task (Fig. 3E). The ES contrast also clearly revealed activity in the posterior cingulate cortex and precuneus, a region that has previously been implicated in decisions to disengage with foraging[33].

Finally, we note that these results were specific to task engagement/disengagement as opposed to response vigor: when we looked at the latter, we were unable to see similar patterns of neural activity to those shown in Figs. 2 and 3 (See Figs. S4, S5 for vigor results). If anything, vigor activity was weaker overall and more transiently related to either decision prompt or after end trial. However, we found a

small cluster of activity related to a future relative increase in vigor (Fig. S5).

## TUS results

Our fMRI analysis identified OE activity in pgACC (Fig. 2C). A study we used in the fMRI analysis also manipulated activity in pgACC using TUS[13] (Fig. 4A) making it possible to assess whether activity was causally responsible for the task engagement level or a consequence of a process that was engendered elsewhere. Thus, we next sought a causal test of pgACC's importance for task engagement. In addition to examining pgACC TUS data, we were also able to examine the impact of TUS in other regions: in the dataset, BF and POp, were also stimulated, and it also include a sham condition[13]. BF is a useful control region because BF activity is associated with the timing of individual actions and BF TUS and cholinergic manipulation (BF is a source of many cholinergic projects) have been shown to alter the timing of individual actions[13,27]. By contrast, POp was not associated with general task engagement/disengagement nor with performance of the specific task and so POp TUS acted as a general control for cortical stimulation. The TUS wave frequency was set to 250 kHz. TUS was applied in 30 ms bursts that were generated every 100 ms for a total period of 40 s. The procedure was then immediately repeated for another 40 s in the same area but in the other hemisphere. All TUS was applied prior to the behavioral task. Sustained TUS trains have previously been shown to exert a sustained impact on neural activity and behavior and therefore make it possible to examine the effect of neural disruption in the absence of any concomitant auditory effects that might be associated with the delivery of each TUS pulse[24,34−37].

To examine the effect TUS had on the time spent disengaged, we classified each timepoint in each session as engaged or disengaged, and calculated the time spent disengaged for each stimulation site as

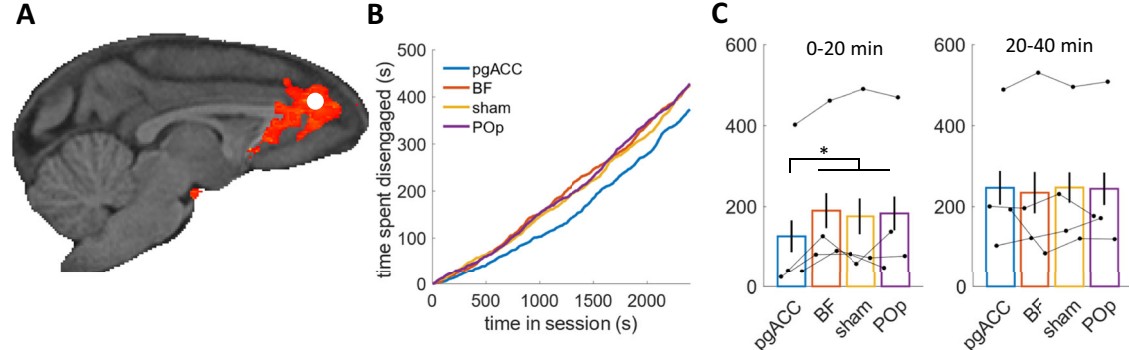

**Fig. 4 | TUS effects on disengagement. A** One of the tasks[13,27] used in our analysis also causally manipulated activity in pgACC using TUS. The stimulation site is shown as a white circle superimposed on the significant OE cluster. Two other regions were also stimulated, and we used data from these TUS sites as controls and also included a sham condition as a further control. **B** The total time spent disengaged by time in the experiment, averaged over sessions and four animals, reveals that after perigenual anterior cingulate cortex (pgACC; blue) stimulation, animals are more engaged early on during the task, compared to the situation after stimulating basal forebrain (BF; red), an anterior parietal region (POp; purple), or sham (yellow). **C** When averaging the time spent disengaged over the first and last 20 min of the task, we find a significant difference between pgACC and the other TUS sites in the first (denoted by *; $\chi^2(1) = 5.27$; $p = 0.022$) but not in the last 20 min. Bars represent condition means, and black dots represent individual subject means, with error bars representing the standard error of the mean. Source data are provided as a Source Data file. The images in **A** were created by the authors using the McLaren template as implemented in the MrCat toolbox. Copyright: 2008-2011, Donald McLaren https://www.gnu.org/licenses/gpl-3.0.en.html.

animals progressed through the session. This analysis revealed a tendency for more frequent early disengagements in the control conditions than after pgACC stimulation, whereas late disengagements appeared equally common throughout all conditions (Fig. 4B). Indeed, when testing for a difference between disengagements after pgACC stimulation compared to other stimulations sides, we found a significant difference in the first 20 min but not in the last 20 min (early: $\chi^2(1) = 5.27$; $p = 0.022$; late: $\chi^2(1) = 0.03$; $p = 0.867$; mixed effects models with random slopes and intercepts for condition by monkey). This effect of more engagement early on can also be observed in each animal individually (Fig. S6).

## Discussion

Task engagement fluctuates throughout daily activity leading to inattention. Ultimately, however, people and animals may give up on a task completely and either remain inactive or pursue an entirely different course of behavior. While the process of error monitoring and subsequent adjustment of behavior has received considerable attention[38–40], less is known about the processes that drive complete task disengagement. This is despite the obvious relevance such mechanisms have to the understanding of apathy—a prominent feature of psychological and neurological illnesses[1]. Although the social demands of the research setting mean that human participants rarely give up on a task completely when they are participating in an experiment, it is not unusual for macaques to move between periods of task disengagement and then re-engagement. In the current investigation we identified such periods and found that they manifested in similar ways across eleven macaques performing four different cognitive tasks in the MRI scanner. When animals were strongly engaged in any task and unlikely to disengage, then a broad region of increased activity spanning several areas, but which was especially prominent in pgACC, was found. Activity was weakest on trials when the animals' task engagement levels collapsed and the monkeys disengaged. The effects were apparent even when we controlled for RT suggesting that pgACC activity was related to task disengagement rather than any change in response timing[13], response control[38,41], or any change in response vigor that might lead to changes in RT[5,42,43]. While vigor and engagement were associated with different behavioral indices and had correspondingly distinguishable relationships with brain activity, some of the effects were adjacent in the brain. However, vigor effects often appeared to be mostly linked with the vigor level on the preceding the current trial (see Fig. S4 and S5) or with increases of vigor that were about to occur (Fig. S4D future vigour—past vigour). By contrast, engagement effects reflected stable patterns of behavior sustained over several trials.

The pattern of activity found in pgACC suggests it is linked to a fundamental process of task engagement that is independent of any particular task identity or specific task feature. This conclusion was reached after observing that the link between pgACC activity and task engagement was found after regressing out any influence that specific task events might have had on neural activity. In fact, for all analyses, we extensively regressed out task parameters to remove all the variance linked to task features and reward history, so that we were able to examine how fluctuations in the residual, activity unrelated to any specific task type was linked to fluctuations in engagement. As such, our findings cannot be attributed to parameters manipulated during the task or satiety and fatigue (we regress out the cumulative reward and the trial number). While we did not examine task-related activity here, this was the focus of previous analyses of all included datasets[13,24–27]. Importantly, each original study shows distinct patterns of neural activity that can be linked to the variables manipulated during each task, which differ from the activity patterns we show here. While we focused on the task independent elements of motivation and engagement there is, of course, a large body of work on motivation, fatigue and apathy based on effort and cost models[44–48]. Future research could potentially combine both approaches (intrinsic/task independent and task driven motivational fluctuations) to get a more comprehensive picture of their interplay.

In all tasks included in the analysis, we could distinguish between activity related to task engagement on a given trial (current engagement; CE)—whether the animal was engaged or disengaged on the trial itself—and the more general state (general engagement; GE) surrounding the trial. Moreover, activity change was not just apparent at the time of responding but it was present and built up over a longer preceding time period. Timecourse analyses revealed elevated signals approximately 15 s before and 15 s after the trial in question. The slowly evolving pgACC signal might reflect the parallel slow evolution of task engagement factors and their independence of specific task events.

Importantly, the corresponding pgACC region of the human brain[49] has been linked to the predisposition to initiate foraging behavior and in determining that the prospect of potential future outcomes mean that it is worth initiating a sequence of behavior

despite potential costs[50,51]. The pgACC is unusual in that it is one of only two cortical regions that project strongly to the striosomal compartment of the basal ganglia, in anterior striatum, which, in turn, is distinguished by a number of anatomical features including projection to the dopaminergic midbrain[9,52,53]. As a result, pgACC is well placed to regulate fundamental aspects of motivated behavior under the control of dopamine[54].

Not only was activity in pgACC predictive of task engagement but TUS-induced alteration of pgACC activity led to consistent patterns of changed task engagement in the four macaques that participated in an additional TUS study. As the TUS stimulation data was part of the original study design[13] we had no control over stimulation sides and could not employ the same stimulation across all tasks or brain sites. While we were unable to examine the impact of stimulating the supracallosal gACC region (Fig. 2D), it was, however, possible to examine the effect of pgACC stimulation because, fortuitously, TUS had been applied to this area in the task investigated by Khalighinejad and colleagues[13]. However, due to only having TUS stimulation in one study, we could not investigate the task general causal impact of pgACC stimulation. After the application of TUS, macaques were less likely to disengage from a task. Normally, when animals were in the control condition, in the first half of a 20 min testing session, macaques disengaged from the task for approximately 3 min. After pgACC TUS, however, animals often worked continually without disengaging or only took a break for approximately 2 min on average. Importantly, the effects were specific to pgACC TUS and were not observed after TUS to two control brain regions. First, similar effects not seen when applying TUS to an anterior parietal control region, POp, in which there was no task-related or task engagement-related activity. Second, and perhaps even more importantly, such effects were not seen when TUS was applied to the cholinergic BF even though it has previously been shown that BF TUS and systemic cholinergic manipulation change the timing of animals' decisions to make individual actions[13,27]. However, while the stimulated pgACC appeared to track our engagement variable, we acknowledge that several factors, such as emotion, energy level and ability to focus, and subjective emotional responses[55] may be directly or indirectly linked to engagement. Future research will need to tease apart.

Relatively few behavioral experiments have focused on the macaque pgACC and previous behavioral analysis approaches have not allowed identification of clear changes in task engagement[13] of the sort that we were able to identify here. However, it has been reported that electrical microstimulation of the macaque pgACC during a cost/benefit decision making task led to fewer decisions to pay higher costs (enduring air puffs) to obtain higher rewards (more juice)[10]. If pgACC is not only responsible for setting the general willingness to endure costs for benefits during choices but also responsible for setting the general level of engagement, then our results and these previous findings can be reconciled. However, it is important to note that TUS is unlikely to recreate patterned excitation of specific neurons that can be induced by microstimulation but rather it may be more likely to disrupt the endogenous activity patterns within a brain region[34,37]. In the rat, optogenetic inhibition of the projections from the homologue of pgACC[56]—often called the prelimbic cortex—to the striosome compartment of the striatum similarly leads rats to be more likely to pay the cost of engaging in a trial in order to obtain a reward[57]. This occurs because pgACC outputs synapse with inhibitory interneurons in the striosome which, in turn, connect with striatal projection neurons. Thus, disrupting pgACC leads to the release of striatal projection neurons from inhibition. As noted, striosomal projection neurons are distinguished by their unique anatomical connections to regions such as the dopaminergic midbrain. In summary, pgACC TUS or pgACC optogenetically mediated inhibition in monkeys and rats respectively make animals more likely to engage in an effortful task to obtain reward or to take a costly action to obtain reward. Both interventions may resemble one another in leading to the release of striatal projection neurons from inhibition and, as a consequence, changes in dopamine levels. While we found an impact of TUS on behaviour, we can only speculate about the physiological mechanism(s) at this stage. Hopefully future research on the physiological impact of TUS and post stimulation fMRI will be able to shed some light onto this question, in particularly why TUS can lead to apparent enhancement of behaviours associated with the stimulated brain regions, despite TUS not mimicking neural activity.

The pgACC region studied here not only has a homologue in rodents but also in humans[49,56]. In humans, coupling between pgACC activity and striatal activity has been linked to disinhibition of effortful choices; first, it was more prominent when the costs of a course of action were high but it was still pursued and second it was more prominent in individuals who were inclined to pursue such courses of action[51]. Individual variation in pgACC activity has also been reported to covary with how influenced each person is by the prospect of future reward despite the need to engage in a sequence of decisions[50]. It also tracks how well people have been performing simple tasks and how they are likely to evaluate their performance[58,59].

The idea that animals make decisions to engage or disengage with one behavior or another or simply to do nothing at all is consistent with a growing body of work on decision making during foraging and their neural correlates[60,61]. It also suggests alternative ways of thinking about situations in which people and animals appear to lack task engagement. In particular, the ES activity in supracallosal cingulate gyrus (area 24, also called mid-cingulate cortex) might normally, in less constrained situations than in the current experiment, lead to sudden deliberate decisions to disengage, rather than simply reflecting slowly waning task engagement. While this is only speculation, it is nonetheless noteworthy that ES specifically activated the supracallosal cingulate gyrus in a region adjacent to one that has been linked to switching and foraging activity in the past in humans, macaques, and rodents[35,50,51,60,62–64] and which is distinct from pgACC. Our results have obvious links to a large body of work on error monitoring[65] and performance lapses[66] in humans that have identified ACC/MCC and pre-SMA as relevant regions for both error monitoring as well as post error adaptation effects. While our lapses are a complete disengagement from the task, not an error per se, the overlap between the anatomical location of the effects reported here and the effects previously reported is intriguing, suggesting common mechanisms may prevent disengagement, maintain engagement with the current task, and mediate performance monitoring for errors and post-error adaptation and return to task performance. Overall, our results suggest slowly drifting fluctuation in engagement where low pgACC activity is linked to low engagement levels and repeated giving up, while sudden and surprising decisions to give up during otherwise high engagement state are triggered by sudden supracallosal gACC activity. The ES was also the only contrast that clearly revealed posterior cingulate cortex and precuneus, a region that has previously been implicated in decisions to disengage with foraging[33], further suggesting that ES might be linked to deliberate decisions to disengagement in a specific trial, as opposed to gradual drifting decline in task engagement.

Overall, our findings not only suggest pgACC mediation of intrinsic variation in task engagement but, more generally, emphasize the multifaceted nature of motivation and task performance. Specifically, we could dissociate task engagement from response speed. However, our ES index suggests that even giving up on a task might not be determined by solely one factor. In fact, in our study, animals might give up because of an overall change in intrinsic task engagement (OE) or because they deliberately, but transiently, want to do something else (ES). We suggest that future work should embrace this complexity.

While we could show task-general and robust neural and behavioural patterns related to task engagement, we do not know what cognitive/emotional or otherwise internal construct is driving

motivation states, as we cannot ask the animal about their subjective experience. It is possible that other fundamental constructs are linked to our pgACC activity in particular, which in turn relate to the motivational state the animal is in, instead of pgACC driving motivation directly. However, whatever such a fundamental construct might be responsible, it appears intimately linked to motivation across tasks.

Importantly, engagement-related activity was not confined to pgACC but was also noticeable in a posterior part of the lateral orbitofrontal sulcus. This region has been identified with credit assignment −the linking of specific choices to specific outcomes[29,34]−but it is also notable that cortex in the same region or nearby is the second cortical region in the macaque, in addition to pgACC, that projects to the striosomal compartment of the stratum, the striatal region that is, in turn, likely to influence the dopaminergic midbrain, and in which stimulation is known to affect cost-benefit decision making[9,52].

While the current study has taken some of the first steps needed to identify the neural mechanisms mediating task engagement, some questions remain unanswered. Notably while pgACC and posterior lateral orbitofrontal sulcus were less active when task disengagement occurred, a more posterior mid-cingulate gyrus region (area 24) was most active during sudden disengagement (Fig. 3). As well as attempting to understanding the key elements that determine the multifaceted relationships between specific task features, task engagement, brain activity and the cellular mechanisms at play in pgACC and beyond, an important future step will be examining the effect of manipulating activity in area 24.

## Methods
### Subjects
13 rhesus macaques (one female) across 17 data sets were included in the four studies considered. All procedures were conducted under licenses from the United Kingdom (UK) Home Office in accordance with the UK Animals (Scientific Procedures) Act 1986 and with the European Union guidelines (EU Directive 2010/63/EU).

### Data collection
The fMRI data were acquired in a horizontal 3 Tesla MRI scanner with a full-size bore using a four-channel, phased-array, receive-only radio-frequency coil in conjunction with a local transmission coil (Windmiller Kolster Inc, Fresno, USA). The animals were head-fixed in a sphinx position in an MRI-compatible chair (Rogue Research, CA). fMRI data were acquired using a gradient-echo T2* echo planar imaging (EPI) sequence with the following parameters: $1.5 \times 1.5 \times 1.5$ mm resolution, 36 axial interleaved slices with no gap, TR of 2280 ms, TE of 30 ms, and 130 volumes per run. Proton-density-weighted images using a gradient-refocused echo (GRE) sequence (TR = 10 ms, TE = 2.52 ms) were acquired as reference for offline image reconstruction.

### Behavioral task-models
We used data from four different tasks[13,24–27]. See the Supplementary Note for descriptions of the four tasks. In all tasks monkeys had to respond to stimuli on screen that were rewarded, while their neural activity was recorded using fMRI. Briefly, Jahn and colleagues (study 1)[25] ran an exploration-exploitation task with different time horizons. In some trials, monkeys had to make one-off choices between two stimuli on screen based on the information presented. In other trials, they had to choose between the same options repeatedly, which enabled them to learn more about the value of the options. Grohn and colleagues (study 2)[26] ran a task with a single option presented on screen. By manipulating the reward associated with the option, as well as the location of the option on the screen, they induced different kinds of surprises. In the study of Bongioanni and colleagues (study 3)[24] monkeys had to choose between two options that varied among two dimensions, reward amount and reward probability. They presented the monkeys with novel stimuli that they had not encountered before

but the value of which they should be able to infer based on previously observed stimuli. Khalighinejad and colleagues (study 4)[13,27] showed monkeys a single stimulus that contained information about the reward amount and the inter-trial-interval length. The longer monkeys waited to respond, the more the reward probability increased, which was also displayed as a feature of the stimulus, but at the price of losing time as the experiment did not have a fixed number of trials but was limited to 40 min. This allowed them to study how monkeys decide when to make a response.

To regress out the effect task manipulations have on engagements/disengagements, we used logistic regressions to control for these effects. For all tasks, we included regressors for the rewards the animals obtained on the previous 5 trials, the current trial number, and the cumulative reward the animals received so far during a session. Additionally, we included task-specific regressors for each task that are based on the models used in the original analyses of the tasks.

For study 1[25], we included a regressor coding for repetition bias (whether the animal has responded on the same side on the previous trial), a regressor coding for the choice horizon (short or long), and a regressor coding for the current choice number within a horizon. Moreover, we used the Bayesian model described by Jahn's and colleagues[25] to estimate the expected reward and the expected uncertainty on each trial. We then included the sum of the expected reward of both stimuli, and the sum of the uncertainty of both stimuli as regressors as well as the absolute difference in expected reward and uncertainty between the two stimuli. Finally, we allowed these 4 latter regressors to vary by horizon as interaction terms.

For study 2[26], we included a regressor coding for whether the stimulus is on the left or the right side of the screen, a regressor coding for whether the stimulus switched sides, and a regressor coding for whether the monkeys received 2 drops of juice on the last trial.

For study 3[24], we included regressors for the absolute additive value difference, the absolute multiplicative value difference, the total additive value, and the total multiplicative value. Additive and multiplicative value here refer to adding or multiplying reward magnitude and probability (further details can be found in the original publication). Moreover, we also included a regressor capturing a repetition bias (responding on the same side as on the previous trial).

For study 4[13,27], we included regressors for the current reward magnitude, the length of the upcoming inter-trial-interval, and the speed of the dots on screen.

All models were run separately for each monkey. For each monkey, we allowed all regressors to also vary as random slopes by session. We then took the difference between the model prediction and observed behavior as our measure of CE.

### Autocorrelation and kernels
To calculate the autocorrelation of our measure of intrinsic task engagement, we shift the timeseries for each session of each monkey by lags from 2 to 10 and compute the correlation for each (we leave out lag = 1 because for some of our experiments, two disengagements cannot occur after each other because of the task design). We then separately average the sessions of each monkey, before finally averaging over monkeys.

To test whether the autocorrelation is significantly larger than 0, we randomly permute the data of each session and repeat the above procedure 10, 000 times on the permuted data. We then determine the $p$ value as the number of times the average autocorrelation over monkeys is smaller than the permuted average. Because we are testing lags from 2 to 10, we use a $p$ value of $0.05/9 = 0.0056$. For RTs we use a $p$ value of $0.05/10 = 0.005$ because we are testing lags 1–10.

To compute the task engagement state, we fitted an exponential kernel to our measure of intrinsic task engagement. Specifically, we found the free parameter $\alpha$ that minimised the squared distance between the function $\alpha(1 - \alpha)^{|d|}N$ and the data, where for

each trial, $d$ indexes all past and future trials of a session, leaving out the current trial, i.e. $d = first\ trial, \ldots, -2, -1, 1, 2, \ldots, last\ trial$, and $N$ is a normalization factor that makes the weights sum up to 1, i.e. $N = \sum \left( \alpha (1-\alpha)^{|d|} \right)^{-1}$. We compute $\alpha$ separately for each of our 4 tasks by finding the $\alpha$ that minimizes this error across all sessions associated with that task. Thus, we overall fit four values of $\alpha$. For study 1[25] and study 4[13,27] we used $d = first\ trial, \ldots, -2, 2, \ldots, last\ trial$—leaving out trials -1 and 1—because disengagements do not occur concurrently because of the task designs.

We then used the fitted value of $\alpha$ to smooth the data, thus obtaining a state estimate on each trial. By using only the half of the kernel that is directed towards the past/future, i.e., $d = first\ trial, \ldots, -2, -1$ and $d = 1, 2, \ldots, last\ trial$, we were also able obtain separate state estimates of the past and future GE, which we used as regressors in the whole brain analysis.

When fitting the kernel to RTs we are only using engaged trials. Therefore, the timeseries are interrupted when a disengagement happens, which also breaks the autocorrelation. For RTs we therefore only use consecutive chunks that are uninterrupted by disengagements to fit the kernel, i.e., we set $d = earliest\ trial\ that\ is\ engaged, \ldots, -2, -1, 1, 2, \ldots, latest\ trial\ that\ is\ enaged$.

### Whole-brain analyses

EPI data were prepared for analysis following a dedicated nonhuman primate fMRI processing pipeline using tools from FSL[67], Advanced Normalization Tools (ANTs)[68], and the Magnetic Resonance Comparative Anatomy Toolbox (MrCat; https://github.com/neuroecology/MrCat).

Like for our behavioral analysis, we also created separate neural regression models for each task. Apart from these task-specific regressors (further outlined below), we also included the same regressors across tasks. For all tasks, we included regressors for the current level of the intrinsic CE (computed as described in the *behavioral task-models* section), the past GE, and the future SM (computed as described in the *autocorrelation and kernels* section). We included all of these regressors twice, once time-locked to the end of the reward delivery of the previous trial, and once time-locked to the onset of the decision-prompt. Moreover, we also included regressors for the trial vigor, and the past and future state vigor, again time-locked both to the end of the previous trial's reward delivery and the decision-prompt. The correlation between these 12 regressors is shown in Fig. S3D.

To compute overall estimates of GE and state vigor, we created contrasts that summed up the past and future GE, and the past and future state vigor. Moreover, to estimate OE we added a contrast that summed up CE and GE, and to estimate ES we added a contrast that subtracted CE and GE. Similar contrasts were included for vigor. Finally, we also included contrasts that subtracted the past and future GE, and the past and future state vigor.

Additionally, we also included some control regressors that were the same for all four tasks. We included intercepts time-locked to the beginning of the reward delivery, the end of the reward delivery, the onset of the decision prompt, and when decisions were made. We also included the current trial number, the cumulative reward so far, and the seconds since the beginning of the experiment, all time-locked to the end of the previous trial's reward-delivery, and to the onset of the decision prompt. Moreover, we also included confound regressors to index head motion and volumes with excessive noise. Motion-related artifacts were captured by including 13 principal components accounting for volume-by-volume magnetic field distortions due to limb and body movements during task performance. Volumes with excessive noise were entirely excluded from the fMRI analysis by including regressors for each flagged volume. Both the 13 principal components and the low-quality volumes were estimated for each session using the MrCat toolbox (https://github.com/neuroecology/

MrCat), as also described in the original publications for each dataset[13,24–27].

Task-specific regressors were based on the models used in the original papers. The regressors we included were:

For study 1[25], we included an intercept time-locked to the onset of the wait-stimulus. We also included regressors for the expected reward of the chosen stimulus, the expected reward of the unchosen stimulus, the uncertainty of the chosen stimulus, and the uncertainty of the unchosen stimulus, all time locked to the wait-stimulus. These quantities were calculated according to the Bayesian model described in the original paper. At decision, we included a regressor for the response side. At the beginning of the reward delivery, we included regressors for the amount of reward received, the expected reward of the chosen stimulus, the expected reward of the unchosen stimulus, the uncertainty of the chosen stimulus, and the uncertainty of the unchosen stimulus, all again according to the Bayesian model. Some sessions also included a horizon manipulation, such that animals had to either make one-off decisions, or decide among the same options multiple times while learning new information about the options throughout. For these sessions, we included a regressor at the decision-prompt whether the trial was a short or a long horizon trial. Furthermore, in some sessions animals received feedback about the reward of the unchosen stimulus, whereas in others they did not. For the sessions that included this feedback, we also included a regressor for the amount of reward of the unchosen option, time-locked to reward delivery.

For study 2[26], we included a regressors for the response side and whether the stimulus had switched sides at decision. At reward delivery, we included regressors for the current reward amount, and the reward amount of the previous 5 trials as separate regressors. We also included a regressor for whether the reward was 2 drops of juice, and a regressor for whether the previous reward was 2 drops of juice. Finally, we also included a regressor for whether the current trial was an error and no reward would be delivered, time-locked to when the reward would otherwise be delivered.

For study 3[24], we included regressors for the absolute additive value difference, the absolute multiplicative value difference, the total additive value, and the total multiplicative value, all time-locked to decision-prompt. These regressors are further described in the original paper. We also included a regressor for the response side at decision, and a regressor for the reward amount at reward delivery.

For study 4[13,27], we included regressors for the current reward magnitude, the upcoming inter-trial-interval duration, and the dot-speed, all time-locked to stimulus presentation. We also included regressors for the last trial's reward amount, and the number of dots on screen when the last trial's response was made, also time-locked to stimulus presentation. At decision, we included a regressor for the number of dots currently on screen. Finally, we included a regressor for the reward amount at reward delivery.

We used a hierarchical GLM approach to combine data from monkeys and sessions: We first fitted each session individually using the appropriate regression model (as described above), and then warped the resulting statistical maps onto the *macaca mulatta* McLaren template in F99[69] as implemented in MrCat. There, on a second hierarchical level, we combined data individually for each monkey using fixed effects and pre-planned contrasts over regressors that were shared across models. Finally, on a third hierarchical level, we combined data from all monkeys using random effects, as implemented in the FLAME 1+2 procedure from FLS[67]. To test for statistical significance, we used a standard cluster-based thresholding criteria of $z > 2.3$ and $p < 0.05$[31].

Analyses were run in FSL's fMRI Expert Analysis Tool (FEAT). Regressors were z-scored and convolved with a hemodynamic response function (HRF), which was modelled as a gamma function (lag = 3, sd = 1.5) convolved with a boxcar function of duration 1 s.

### ROI analyses and timecourses

To define ROIs, we calculated the overlap between the cluster-corrected t-statistic map from the whole-brain analysis and anatomically defined regions based on an atlas[32], which we dilated with a kernel of 3x3x3 voxels. We then warped these ROIs into session-space using the nonlinear deformation field.

To visualise the BOLD timecourse of a regressor we re-ran the convolutional whole-brain analysis for each session of each monkey in FEAT, leaving out the 12 regressors of interest we described above but including all other task-relevant and nuisance regressors. We then extracted the average residual of this whole-brain analysis from each ROI. Next, we upsample the timecourse by a factor of 10 using spline interpolation. Because we are interested in temporally extended effects of task engagement, we then smooth the upsampled timecourse with a moving average filter of 5 s.

### TUS stimulation and analysis

TUS stimulation was conducted with a single-element ultrasound transducer (H115-MR, diameter 64 mm, Sonic Concept, Bothell, WA, USA) with region-specific coupling cones filled with degassed water and sealed with a latex membrane (Durex). The ultrasound wave frequency was set to the 250 kHz resonance frequency and 30 ms bursts of ultrasound were generated every 100 ms (duty cycle 30%) with a digital function generator (Handyscope HS5, TiePie engineering, Sneek, the Netherlands). Overall, the stimulation lasted for 40 s. A 75-watt amplifier (75A250A, Amplifier Research, Souderton, PA) was used to deliver the required power to the transducer. For further details, see ref. 13.

To calculate the time spent disengaged, we classified each trial in each session as engaged or disengaged in the same way we did for the data sets for the behavioral and fMRI analysis. We then calculated the total time spent disengaged for each session, and tested whether there was a significant difference between the sessions in which pgACC was stimulated or the control conditions (BF, POp, or sham stimulation). In this model, we also included a random intercept for each animal to control for different baseline effects and a random slope for whether pgACC or a control side was stimulated.

To visualize where in a session differences between conditions emerged, we also calculated the cumulative sum of the time spent disengaged for each second of each session, and then averaged this sum over sessions for each condition.

### Reporting summary

Further information on research design is available in the Nature Portfolio Reporting Summary linked to this article.

## Data availability

All datasets used in this study have already previously been published. Please contact the corresponding authors of the original publications for access to the raw datasets. The processed data are available at https://doi.org/10.5281/zenodo.10960864. Source data are provided with this paper.

## Code availability

Code to replicate the analyses and figures shown in this paper can be found at https://doi.org/10.5281/zenodo.10960864

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

## Acknowledgements

Funding for this work was provided by Medical Research Council (https://www.ukri.org/councils/mrc/) grants MR/P024955/1 (MFSR, J.S., and N.K.), G0902373 (MFSR), MR/K501256/1 (J.G.), MR/N013468/1 (J.G.), Wellcome Trust (https://wellcome.ac.uk/) grants 203139/Z/16/Z (MFSR); WT100973AIA (MFSR); WT101092MA (MFSR and J.S.), 105651/Z/14/Z (J.S.), St John's College, University of Oxford (https://www.sjc.ox.ac.uk/) (J.G.), the Biotechnology Biological Sciences Research Council (https://

www.ukri.org/councils/bbsrc/) grant BB/R010803/1 (N.K.), the European Research Council (https://erc.europa.eu/) grant FORAGINGCORTEX, project number 101076247 (N.K.), the Economic and Social Research Council (https://www.ukri.org/councils/esrc/) grant ES/J500112/1 (A.B.), and a studentship from the Paris Descartes University doctoral and a mobility grant (C.J.). The funders had no role in study design, data collection and analysis, decision to publish, or preparation of the manuscript. Views and opinions expressed are those of the author only and do not necessarily reflect those of the funders. Neither the European Union nor the granting authority can be held responsible for them.

## Author contributions

NIM.K., C.J., A.B., U.S., and J.S. provided behavioral and neural data. J.G. analyzed the data. NIL.K. and M.R. supervised the project. J.G., NIL.K., and M.R. wrote the paper. All authors commented on the draft of the paper.

## Competing interests

The authors declare no competing interests.
