## [Peer Review File · Nature Communications]

General mechanisms of task engagement in the primate frontal cortexREVIEWER COMMENTS

Reviewer #1 (Remarks to the Author):

The authors investigated the intrinsic component of engagement in animals by examining the behavior of macaques in four different tasks while recording fMRI signals. The authors tried to identify patterns of behavior predictive of task disengagement by developing models to capture task engagement and link it to neural activity. Their findings may reveal that activity in the perigenual anterior cingulate cortex (pgACC) was associated with task engagement across all four tasks. The researchers demonstrated the effect of pgACC on task engagement by altering it with transcranial ultrasound stimulation. Despite the significant and important findings, some concerns remain, particularly regarding the analysis procedures.

Major comments:

(1) The definition of "disengagement":

I have some concerns about the way the authors defined "disengagement." While the authors chose "log RTs" to model engagement using a linear model, I am not convinced that this is the most appropriate way to define and measure disengagement. Some questions come to mind:

(1-1) Why did the authors choose "RT" as the indicator of engagement and disengagement? It would be helpful for the authors to provide a rationale for this decision.

(1-2) Depending on the internal variable selection for the model, the ability of the model to capture disengagement could vary. Therefore, the authors should explain how well their model captures disengagement. Additionally, autocorrelation with the lag may not be sufficient to fully capture disengagement, and other metrics could be considered to enhance the definition.

(1-3) Figure 1D displays a temporal sequence of disengagement, represented by black dots. I wonder why the authors did not utilize this sequence to explicitly characterize the level of disengagement.

(1-4) If the authors can use other behavioral features to quantify disengagement, it would be beneficial to compare the quality of the regression model for each task.

Overall, I recommend that the authors provide additional clarification and justification for their choice of the "RT" metric to define engagement and disengagement, as well as any other metrics they may choose to utilize. Additionally, it would be helpful to provide an evaluation of the performance of the model in capturing disengagement using the available behavioral data.

(2) Weak reasons to use four different tasks.

Firstly, each task had different requirements, resulting in varying degrees of engagement from the monkeys. This indicates that the frequency and triggers of disengagement would be different across the

tasks. Secondly, the quality and correlation between the model residual and behavioral disengagements also varied among the tasks. Additionally, the t-statistics results in Figure 3 differed significantly among the tasks. However, it is unclear if the authors have provided a clear justification for using a consistent definition of disengagement across all four tasks.

(3) The definition of "engagement shift".

In Figure 2, I was unable to determine the meaning of each color. A color bar indicating the level and degree of activation would be helpful in each panel. Furthermore, how did the authors choose the white line to be $z > 2.3$? The definition of "engagement shift" as GE-CE seems too abrupt for me to understand. What is the significance of this measure in relation to "disengagement," which is the primary concept of this article? In general, when the researchers interpret the difference in activity between two conditions, the two conditions needed to be somehow comparable. From this viewpoint, the current interpretation of "engagement shift" might need some more explanation or careful definition.

(4) Weakness of TUS experiments.

While the results of the TUS experiment that demonstrated the causal involvement of the pgACC in engagement are impressive, there are some limitations to consider. For instance, the criteria used to select the tasks for the experiment are not clear. Additionally, the authors claim that the pgACC is the region involved in task engagement, as stated in lines 29 and 30 where they note "Across all tasks, we identified common patterns of neural activity linked to impending task disengagement in mid-cingulate gyrus. By contrast, activity centered in perigenual anterior cingulate cortex (pgACC) was associated with maintenance of task performance." However, the TUS experiments were only conducted on one task, making this statement potentially confusing. The authors may want to clearly separate the common pattern and the potentially different TUS effect across tasks.

Minor comments:

(1) In Fig. 2, please illustrate not only the areas positively related to task disengagement.

(2) In Fig. 3, why this analysis method shows a strong correlation when using the results of Bongioanni et al.?

(3) Regarding the Table of MRI results, please clearly indicate which brain regions correspond to each Z-value point.

Reviewer #2 (Remarks to the Author):

Fluctuations in the level of engagement to a task are commonly observed in daily life. The authors argue that, due to the social demands of the research setting, human participants rarely give up on a task completely when they are participating in an experiment, it is not unusual for macaques to move between periods of task disengagement and then re-engagement. Macaques thus appear to be a good model to investigate this phenomenon. In the present study, the authors focus on general mechanisms of task engagement and disengagement across four different tasks while recording brain activity using fMRI. The authors

constructed separate logistic regression models for each of the four tasks. Each model contained task specific regressors, they also accounted for intrinsic motivation that has previously been linked to satiation such as cumulative reward, or time spent on a task. By regressing out these effects related to task structure and task events, the authors claim the residuals contain the fluctuations in task engagement that are intrinsic as opposed to those that are due to extrinsic factors related to task structure and task events. The authors also constructed 3 variables, “current engagement”, “general engagement” and “overall engagement”, to search brain correlates. The authors argue that such model-derived estimates capture aspects of task engagement not previously reported in the literature. Finally, the authors analyzed the data acquired after acute transcranial ultrasound stimulation (TUS) of pgACC, a region identified to be involved in overall engagement, and compared results from TUS targets in control regions.

In general, this is an interesting study that employs a model-based approach to look at a phenomenon that are commonly observed but have been rarely investigated. The manuscript is well written and easy to follow despite of the complicated methods and procedures involved. Nevertheless, I have quite a few questions, both on the technical aspects and on the interpretation.

Major Points:

1. One fundamental assumption in this study is that, by constructing task specific regressors and additional intrinsic motivational regressors linked to satiation, and regressing out these effects related to task structure and task events, the residuals contain the fluctuations in task engagement that are intrinsic during the experimental sessions. The reviewer would imagine motion-related artifacts could be a major confound in the residues. It is well established that body motion outside the FOV induces susceptibility changes in the brain, and thus signal changes, even with headpost implantation on the animals' head that presumably have minimized physical head displacement. There could be plenty of signal fluctuations related to body motion during task execution, which could, of course, generate auto-correlation in the time course, at least in the awake monkey data that reviewer previously looked at. Obviously, the motion is specific to experimental setup and task designs. Persistent autocorrelation in the residuals was used as the major evidence to support the existence of “current engagement”.

Somewhat to my surprise, the authors did not mention motion issue in the data analysis. How were potential confounding factors from motion handled?

2. The interpretation of the pgACC TUS data. Unlike stimulating sensory or motor systems, stimulating the cingulate cortex could elicit autonomic reaction and complex subjective responses. Such effects are difficult to assess in animals, but have been reported in humans. In this regard, I would like to point out a paper by Parvizi and Greicius et al. 2013 (The Will to Persevere Induced by Electrical Stimulation of the Human Cingulate Gyrus). Thus, it would appear questionable to attribute the effects observed after acute TUS as simply a change in engagement.

3. More generally, are “engagement” and “disengagement” fundamental constructs of animal behavior or a behavioral expression of some more fundamental constructs? In this regard, is it possible to fit the observed finding in a framework of default mode network and attention network?

Specific points:

1) Figure 1 shows CE, GE and general engagement kernel, do the black dots in Fig. 1D indicate moments of disengagement? It would be helpful to also illustrate the temporal relationship between the negative peaks in CE as shown in Fig. 1D and moments when cues/tasks were presented and executed. Also there is no time scales in the trials (x-axis).

2) Kernel derivation: from the method description, it is difficult to understand the criteria (if any) to determine alpha in the kernel function. Is it task specific, is it also run-specific? How was the data from multiple runs combined?

3) In the data shown in Fig. 2, there are many spotty clusters in the activation maps, which lead to classical questions about false positive/negative and thresholding in fMRI. Given the small number of monkeys involved in each task, a table should be added to summarize number of trials, runs, run duration, number of disengagement events as indicated in the black dots of Fig. 1D. Data like these could indicate the robustness of the statistics.

4) TUS effects on disengagement. Of the 4 monkeys received TUS (Fig. S5), only monkey 1 appears substantially deviate from sham stimulation; monkey 3 is basically non-differentiable. This begs the questions of TUS data interpretation.

5) Trial vigor: response times were used to index trial vigor. Results from this analysis was used as a control to support the main claim. However, by careful examination of Fig. S4, cerebellum seems heavily involved in “trial vigor”, is there any explanation for this finding?

6) A brief explanation of the 4 experimental designs should be warranted in the supplemental such that the potential readers do not have to go back previous publications to look for details.

Reviewer #3 (Remarks to the Author):

The manuscript addresses brain activity associated with (dis)engagement from task in four fMRI studies carried out in macaque monkeys. Disengagement was defined as trials with no response or (in three tasks) trials with very long reaction times. Task-related activity and other factors likely reflecting fatigue, satiety, task history, RT variability etc. were regressed out. Maintenance of task performance was associated with activity in the pregenual cingulate cortex. Midcingulate activity was associated with impending task disengagement. Transcranial ultrasound stimulation of the pgACC reduced the occurrence of task disengagement in the first 20 min after sonication in an offline stimulation protocol. This suggests that pgACC is causally related to task engagement. Importantly sonication of control regions, particularly the basal forebrain, did not have reliable effects on the engagement/disengagement patterns.

The authors address an interesting and timely topic that is difficult to study in humans. The methodological approach is innovative and convincing, given that the results are based on four different tasks. The results may open a new path of research, which should be related to other research topics such as fatigue, lapses of attention, mental effort and mind wandering.

I have a few minor issues and suggestions that may be helpful in a revision of the manuscript.

1. The abstract suggests: "We identified consistent patterns of behavior predictive of impending task disengagement". I was wondering which behavioral patterns the authors relate to. The autocorrelation analysis suggests that engagement and disengagement phases are clustered. But are there any other parameters that might predict an impending cluster of disengagement or sudden disengagements (e.g. RT slowing, increased RT variance, errors)?
2. I found the direction of the fMRI and TUS findings in pgACC somewhat intriguing. While fMRI data suggest an association of increased pgACC activity with task engagement (and a number of neuroimaging findings consistent with this are cited in the ms) TUS seems to reduce task disengagements. As stated in the MS it seems unlikely that ultrasound effects mimic physiological activity such that I would have expected a detrimental effect of TUS in pgACC on task engagement. Some more elaboration on these effects might help readers less familiar with the methods interpreting the findings.
3. A number of previous papers might be relevant for this study and the authors might consider discussing them:

- Magno E, Foxe JJ, Molholm S, Robertson IH, Garavan H (2006) The anterior cingulate and error avoidance. *J Neurosci* 26:4769-4773.

This study might support the finding that midcingulate activity might be related to sudden shifts in engagement. Here, midcingulate and anterior insular activity was related to "escaping" trials that were expected to be particularly effortful and perhaps less likely to be successful.

- Eichele T, Debener S, Calhoun VD, Specht K, Engel AK, Hugdahl K, von Cramon DY, Ullsperger M (2008) Prediction of human errors by maladaptive changes in event-related brain networks. *Proc Natl Acad Sci U S A* 105:6173-6178.

Here slow and systematic changes in brain activity preceding erroneous responses are shown (increasing activity not in pgACC but in posterior cingulate/precuneus and decreasing task-related lateral and medial frontal activity) which was interpreted as accompanying gradual disengagement from task.

There are also a number of EEG studies showing systematic changes of brain activity predicting errors, but they are less informative regarding the underlying functional neuroanatomy.

- Weissman DH, Roberts KC, Visscher KM, Woldorff MG (2006) The neural bases of momentary lapses in attention. *Nat Neurosci* 9:971-978. and Chee MW, Tan JC, Zheng H, Parimal S, Weissman DH, Zagorodnov V, Dinges DF (2008) Lapsing during sleep deprivation is associated with distributed changes in brain activation. *J Neurosci* 28:5519-5528.

The definition of lapses of attention is quite similar to the RT-based definition of disengagement used in the present manuscript.

4. While the authors suggest that the vigor-related analysis does not show activity overlapping with the (dis)engagement-related activity, I am not fully convinced when comparing the figures. What is called frontopolar cortex in Fig S3A seems to at least partly overlap with the activity around the pgACC cluster in the main analysis.

5. What could be the reason for the much stronger t-statistics in the 4th study (Fig 3, Bongioanni et al.)?

6. While regressing out fatigue effects does make sense in the present study, the topics fatigue, task engagement, and apathy are related to each other. I would appreciate a discussion on the interaction between fatigue and task engagement. Perhaps the results by M. Apps might be of relevance as well.

7. The fMRI setup precludes any investigation on what the monkeys might do when disengaging from task. Are there any systematic reports in the literature what macaques would engage in when interrupting task-engagement? Could disengagement from one task also be driven by the impression that another task has become more important at this point? This would be substantially different from a fatigue-driven disengagement or from mind wandering.

Firstly, we are very happy to hear that all reviewers found the manuscript interesting, important and innovative. We also wanted to thank all three reviewers for reading our manuscript so thoroughly, and for the helpful comments and suggestions they provided. Below we give a detailed point-by-point reply.

The reviewers' comments are shown in *italics*, with our replies in blue. Edits to the manuscript are highlighted in red.

Reviewer

#1

The authors investigated the intrinsic component of engagement in animals by examining the behavior of macaques in four different tasks while recording fMRI signals. The authors tried to identify patterns of behavior predictive of task disengagement by developing models to capture task engagement and link it to neural activity. Their findings may reveal that activity in the perigenual anterior cingulate cortex (pgACC) was associated with task engagement across all four tasks. The researchers demonstrated the effect of pgACC on task engagement by altering it with transcranial ultrasound stimulation. Despite the significant and important findings, some concerns remain, particularly regarding the analysis procedures.

Thank you for reviewing our manuscript. Upon reading your comments, we realised that we were not as clear about how and why we conducted our analysis as we thought we were. We hope that the revised manuscript provides a better explanation for our rationale for some of our analysis choices. We believe this has strengthened our manuscript considerably and want to thank the reviewer for this.

Major comments:

(1) The definition of "disengagement":

I have some concerns about the way the authors defined "disengagement." While the authors chose "log RTs" to model engagement using a linear model, I am not convinced that this is the most appropriate way to define and measure disengagement. Some questions come to mind:

(1-1) Why did the authors choose "RT" as the indicator of engagement and disengagement? It would be helpful for the authors to provide a rationale for this decision.

The idea for writing this paper came about when we observed that our animals sometimes completely disengaged from the tasks they were performing, and that this happened across all kinds of different tasks. The animals would pause for a bit (in the range of a couple of seconds to minutes) before resuming the task. As we were scanning these animals using fMRI as the disengagements happened, we became interested in the neural patterns associated with the decision to disengage or engage with a task. Because we observed disengagements in our animals regardless of task, we thought that at least some of their disengagements were likely driven by something external to the task at hand, and we wanted to isolate and study this component of disengagement/ engagement. The only way to quantify when an animal disengaged from a task was to look at their responses; if they did not respond for a while, then they must be disengaged. Thus, we used response times that were long enough that they

could not feasibly come from an animal that is “on task” (i.e., engaged with the experiment) and classified all responses above a high threshold as disengaged. We deliberately chose to binarize response times using a threshold (as illustrated in Fig 1A) as we were conceptualising disengagements as all or none decisions rather than continuous fluctuation of reaction times. In short, we set the threshold at a high level to ensure that any disengagement identified were not just occasions when animals were responding sluggishly but still on task. Our approach is thus conservative and ensures that we avoid false positively labelling of trials as disengagements when they were performed a little slower than usual. Interestingly this also allowed us to contrast such disengagement choices with “vigor”, which we defined as fluctuation of (log) response times while the animals were still responding reasonably quickly.

We note that there are of course many ways to think about disengagement as it is a multifaceted problem. For example, we could have tried to define disengagement through error rates or other behavioural features. However, our method was chosen because it is generalizable across tasks and could be employed even with those tasks that were so easy that animals barely made any errors. Additionally, errors can reflect many other cognitive processes unrelated to disengagement (especially in our learning tasks as here “errors” can indicate exploration).

We now more clearly spell out this rationale in our revised manuscript. The updated first paragraph of the Results section now reads:

“For the purpose of our analysis, we define disengagements as responding after 3 s or later, or not responding at all during a trial, i.e. the trial “timed out” before a response was made. However, for one of our tasks that incentivized late responses (Khalighinejad et al., 2020, 2022), we only counted trials as disengaged where the animal did not respond at all (see Fig S1 for details for all tasks). We binarized trials into ones where the animals are engaged or disengaged (Fig 1A). **This definition of disengagements conceptualizes behavior as all or none events which we can contrast with a continuous measure of response vigor i.e. when the animals remain on task but respond more or less rapidly (see below). While other definitions of disengagement might be possible (e.g. by looking at decision errors), those would have not been applicable in our tasks due to the large variations in difficulty across task and because errors can occur during learning as well as when there is disengagement tasks. By applying our response time-based definition, we can consistently classify disengagements across a range of diverse tasks and capture the intrinsic, task-independent nature of these events. Our threshold of 3 s was chosen to ensure that on trials that were classified as disengagements, the animals made the decision to disengage rather than responding sluggishly while still being on-task”**

(1-2) Depending on the internal variable selection for the model, the ability of the model to capture disengagement could vary. Therefore, the authors should explain how well their model captures disengagement. Additionally, autocorrelation with the lag may not be sufficient to fully capture disengagement, and other metrics could be considered to enhance the definition.

The reviewer is correct that more complex models might have somewhat higher predictive power than our trial wise disengagement regressor. However, part of the

reason we used an autoregressive model is for its interpretability: it reflects a slowly fluctuating internal variable, which we assumed intrinsic motivation might well be and as was reaffirmed by our neural analyses.

We also agree with the reviewer that autocorrelation may only capture part of disengagement. However, the aim of using the autocorrelation across trials to construct our measure of general engagement was not to find the model that was best at explaining individual disengagement trials but rather to obtain a regressor for our fMRI analysis that captured the engaged state on a trial, and that was interpretable when regressed against BOLD. Conceptually, the general engagement level is high if there is low temporal clustering of disengagements, and low if there is high temporal clustering of disengagements (illustrated in Fig 1D). This allows for interpreting the contrasts we use in our fMRI analysis (also see our answer to point 3 of the reviewer regarding 'engagement shifts' below), which we prioritised over explaining the most behavioural variance.

We have added a sentence to the results section that points towards this rationale:

“To this end, we fit exponential kernels to the residual fluctuations (Fig 1C right shows the fitted kernel for each of the four tasks). These kernels capture the extent to which task engagement on a trial, as indexed by the residual fluctuations, is related to task engagement on preceding and following trials. Smoothing the residual fluctuations (CE; orange line in Fig.1D; shown after normalizing) by these kernels allows us to obtain an estimate of a continuously varying GE (blue line in Fig.1D; shown after normalizing) on each trial. **We construct GE this way to obtain an interpretable regressor we can use in our fMRI analyses.**”

The reviewer might also be wondering how much variance in engagement is actually explained by our task models. In the plot below, which we added to the supplementary materials, we now show histograms of what we call “current engagement” in the manuscript split up by task. Current engagement is the residual of the logistic regression models (i.e. the task-specific models appropriate for the tasks used in the four experimental data sets) we used to identify periods of task where the monkeys stopped responding. Trials on which the animals were engaged are shown in blue, and trials on which the animals were disengaged are shown in orange. As can be seen, the histograms do not appear to show vastly different patterns across tasks, suggesting that the ability of the model to predict disengagements is similar for each dataset. Noticeably, our task-model does not capture all variance in behaviour, which is why, as we argue throughout the manuscript, the residuals can be used as an estimate of intrinsic engagement on each trial.

The figure legend reads:

“*Figure S2. Density of current engagement (CE) split by whether the animals were engaged (blue) or disengaged (orange) on a trial. The density of CE for engaged and disengaged trials are normalized separately. For all trials, a value close to 0 indicates that a trial was well predicted by the task-specific regression model. Because CE is the residual of a logistic regression model that predicts engagement with the task, a value closer to 1 indicates that the model predicted a disengagement, but the animal did not disengage. By contrast, a value close to -1 indicates that that the model predicted the animal would be engaged but the animal disengaged on this trial.*”

(1-3) Figure 1D displays a temporal sequence of disengagement, represented by black dots. I wonder why the authors did not utilize this sequence to explicitly characterize the level of disengagement.

We assume the reviewer is thinking of somehow adding up the number of disengagements or their clustering? If that is the case, we think that our autoregressive model is a good way to capture these sequences/patterns/clusters of disengagement as they simply filter all the black dots with a recency weighted filter to get a state prediction, except that we also take into account that some of those black dots might be a direct consequence of task features (e.g. the animal is annoyed at not getting rewarded), which is why we regress out those features first. By regressing out task-related variables such as stimulus location, reward and trial number, we are able to construct a regressor that captures the disengagement that is not explained by the task. This can also be seen in the histograms shown above in our reply to question (1-2)—while some disengagements are not well explained by the task-model (current engagement close to -1), some of them are better explained by the task-model (current engagement closer to 0). By using those “post task model” regressor for our model and neural analysis rather than the disengagement trials directly, we wanted to avoid obvious impacts of reward and other environment features that might drive disengagements. Additionally, we also account for task-effects in our neural analysis by including the full task-model as regressors there as well.

(1-4) If the authors can use other behavioral features to quantify disengagement, it

would be beneficial to compare the quality of the regression model for each task. Overall, I recommend that the authors provide additional clarification and justification for their choice of the "RT" metric to define engagement and disengagement, as well as any other metrics they may choose to utilize. Additionally, it would be helpful to provide an evaluation of the performance of the model in capturing disengagement using the available behavioral data.

We hope that the additional supplementary figure S2, which we also pasted above in reply to question (1-2) can be used to assess the quality of the regression model when fitted to each task. As can be seen by visually inspecting the histograms, the model fits seem to not vastly differ between tasks. We also hope that our response to question (1-1) regarding the use of the "RT" metric provides additional justification for the choices we made in analysing our datasets. Apart from disengagements and RTs, we unfortunately do not have any other behavioral readout that is present and conceptually has the same interpretation in all four tasks. Therefore, we are unfortunately unable to relate our regression models to other behavioral features.

(2) Weak reasons to use four different tasks. Firstly, each task had different requirements, resulting in varying degrees of engagement from the monkeys. This indicates that the frequency and triggers of disengagement would be different across the tasks. Secondly, the quality and correlation between the model residual and behavioral disengagements also varied among the tasks. Additionally, the t-statistics results in Figure 3 differed significantly among the tasks. However, it is unclear if the authors have provided a clear justification for using a consistent definition of disengagement across all four tasks.

We would like to apologise for not better articulating our reasons for using four tasks. However, using different task with different requirements was a deliberate choice. We believe that using several tasks has several key strengths. Firstly, by showing that our model works and shows consistent results across tasks, we can illustrate a task-general mechanism for intrinsic motivation further supporting our claim that this motivational element exists beyond the narrowly defined demands of one specific task. Furthermore, it serves as a type of replication of the findings, showing their robustness. While we do agree with the reviewer that the t-statistics do vary between the tasks and that they have identified likely causes in the differences in demands between the tasks, we believe the fact that we can see the same patterns throughout to be a strength, particularly as the tasks the animals performed were chosen to be quite distinct. This is also why we used a consistent definition of disengagement between tasks, as this shows generalizability across our tasks and can hopefully usefully allow others to employ a similar idea to look at such signals in other macaque or even rodent tasks with sufficient RT variability.

We have now added the following text to the first paragraph of the Behavioral Results section to make this clearer:

Using the same analysis approach across tasks is essential for generalizability but also means we had to find a definition of disengagement that works across studies. Thus, while there might be some adjustment in the behavioral definition that could be made if we had only analyzed a single task, we employed an approach with the merit of general applicability; while we might have failed to detect task-specific motivational factors, the approach achieves the aim of identifying neural processes common to many situations.

There is, however, of course a lot of interesting differences between tasks that future studies could explore with more systematic sampling of possible task space (e.g. learning vs non-learning or easy vs difficult tasks).

(3) *The definition of "engagement shift". In Figure 2, I was unable to determine the meaning of each color. A color bar indicating the level and degree of activation would be helpful in each panel. Furthermore, how did the authors choose the white line to be $z > 2.3$? The definition of "engagement shift" as GE-CE seems too abrupt for me to understand. What is the significance of this measure in relation to "disengagement," which is the primary concept of this article? In general, when the researchers interpret the difference in activity between two conditions, the two conditions needed to be somehow comparable. From this viewpoint, the current interpretation of "engagement shift" might need some more explanation or careful definition.*

Apologies for the lack of explanation for this analysis. To clarify, engagement shifts are defined through a contrast between two regressors, not conditions. Thus, the contrast essentially identifies activity that is largest when the animal disengages (disengagement = -CE) despite being in a period of the task where they are generally highly engaged (GE). In other words, they shift suddenly between being engaged to being disengaged, and back. We refer to this as an "engagement shift" but could equally well, perhaps, have also called it an unexpected or sudden engagement lapse. Thus, it is a contrast that identifies trials during which CE and GE make differing predictions (one is low and the other one is high). $Z = 2.3$ was used as a threshold as it is the most common standard in neuroimaging studies, balancing false positives and negatives. It is thus the default setting in the FSL toolbox that we employed and purposefully avoided adjusting. $Z = 2.3$ corresponds to a p-value of 0.01 when the degrees of freedom are large, and before multiple comparison corrections are applied.

We have updated Figure 2 to now also include colorbars. We also now clarify this result and its meaning in the MS:

We also looked for effects of ES, i.e. the difference between GE and CE (Fig 2D). Such activity was prominent when animals disengaged on the current trial while otherwise having been in an engaged state and likely to soon return again to an engaged state. In other words, the analysis identifies 'surprising' disengagements, where the disengagement is not preceded or followed by other disengagements; **or conversely engagement in a disengaged state. It thus identifies trials where our GE and CE indexes are opposed.**

(4) *Weakness of TUS experiments. While the results of the TUS experiment that demonstrated the causal involvement of*

the pgACC in engagement are impressive, there are some limitations to consider. For instance, the criteria used to select the tasks for the experiment are not clear. Additionally, the authors claim that the pgACC is the region involved in task engagement, as stated in lines 29 and 30 where they note "Across all tasks, we identified common patterns of neural activity linked to impending task disengagement in mid-cingulate gyrus. By contrast, activity centered in perigenual anterior cingulate cortex (pgACC) was associated with maintenance of task performance." However, the TUS experiments were only conducted on one task, making this statement potentially confusing. The authors may want to clearly separate the common pattern and the potentially different TUS effect across tasks.

Inclusion of different experiments was primarily practical. We wanted to have a diverse set of studies with different tasks but always with sufficient variability in RT to make analyses possible. Luckily in one of the studies we identified TUS had been conducted across several regions. Even more fortunately, it was done, in part, in relevant brain regions, which is why we choose to analyse the behavioural impact of this stimulation. As it was pre-existing data we had no control over the TUS stimulation location and therefore could not repeat stimulation across tasks.

We apologize for our previously unintended misleading statement. We have now clarified the manuscript text that TUS was only analysed for one of the four studies as it had been conducted in several relevant brain regions:

The last sentence of the Abstract now reads:

Moreover, we showed pgACC activity had a causal link to task engagement; **in one of our tasks**, transcranial ultrasound stimulation of pgACC, but not of control regions, changed task engagement/disengagement patterns.

We also now further clarify the limits of our TUS results in the discussion:

Not only was activity in pgACC predictive of task engagement but TUS-induced alteration of pgACC activity led to consistent patterns of changed task engagement in the four macaques that participated in an additional TUS study. **As the TUS stimulation data was part of the original study design (Khalighinejad et al., 2020) we had no control over stimulation sides and could not employ the same stimulation across all tasks or brain sites. While we were unable to examine the impact of stimulating the supracallosal gACC region (Figure 2D), it was, however, possible to examine the effect of pgACC stimulation because, fortuitously, transcranial ultrasound stimulation had been applied to this area in the task investigated by Khalighinejad and colleagues (Khalighinejad et al., 2020). However, due to only having TUS stimulation in one study, we could not investigate the task general causal impact of pgACC stimulation.**

Minor

comments:

(1) *In Fig. 2, please illustrate not only the areas positively related to task disengagement.*

Fig 2 shows significant results that survived a cluster-correction procedure. Activity that was directly related to negative engagement (i.e. disengagement) did not survive cluster-correction at the threshold we used. As such, we believe it would not be

appropriate to illustrate this non-significant result in such a way. However, the full z-statistic maps associated with the non-significant negative results are provided as part of the extended data, and can thus be viewed by an interested reader.

(2) *In Fig. 3, why this analysis method shows a strong correlation when using the results of Bongioanni et al.?*

First, to clear up any potential confusion, we would like to clarify that the results we plot split up by tasks in Fig 3 do not intend to show the 'correlation strength' directly but the t-statistics for contrasts in our whole-brain analysis within an ROI, which can be impacted by several factors beyond correlational strength. As to why the t-statistic for one of the contrasts is the strongest for Bongioanni et al., we unfortunately do not have enough data to conduct any proper analyses, but we would like to point out that the behavioural autocorrelation we examine is the strongest for that task (Fig 1C). This autocorrelation is then used to construct the 'general engagement' contrast we use in the whole-brain analysis, and it is for this 'general engagement' contrast that we observe the strong effect for Bongioanni et al. As such, it seems plausible that the strong effect is simply due to our model working best for that task. Additionally, as can be seen in Figure S1, the proportion of disengaged trials is larger for the two monkeys that participated in Bongioanni et al. (around 20%) than in other tasks (where it is around 10%). This might also lead to the neural signals appearing stronger in these two animals. However, while the effect is the strongest for Bongioanni et al., the effect is, overall, in the same direction for all 3 other task as well.

(3) *Regarding the Table of MRI results, please clearly indicate which brain regions correspond to each Z-value point.*

We have updated the tables in the supplementary materials to now include an anatomical description for each sub-peak.

Reviewer #2 (Remarks to the Author):

Fluctuations in the level of engagement to a task are commonly observed in daily life. The authors argue that, due to the social demands of the research setting, human participants rarely give up on a task completely when they are participating in an experiment, it is not unusual for macaques to move between periods of task disengagement and then re-engagement. Macaques thus appear to be a good model to investigate this phenomenon. In the present study, the authors focus on general mechanisms of task engagement and disengagement across four different tasks while recording brain activity using fMRI. The authors constructed separate logistic regression models for each of the four tasks. Each model contained task specific regressors, they also accounted for intrinsic motivation that has previously been linked to satiation such as cumulative reward, or time spent on a task. By regressing out these effects related to task structure and task events, the authors claim the residuals contain the fluctuations in task engagement that are intrinsic as opposed to those that are due to extrinsic factors related to task structure and task events. The authors also

constructed 3 variables, “current engagement”, “general engagement” and “overall engagement”, to search brain correlates. The authors argue that such model-derived estimates capture aspects of task engagement not previously reported in the literature. Finally, the authors analyzed the data acquired after acute transcranial ultrasound stimulation (TUS) of pgACC, a region identified to be involved in overall engagement, and compared results from TUS targets in control regions.

In general, this is an interesting study that employs a model-based approach to look at a phenomenon that are commonly observed but have been rarely investigated. The manuscript is well written and easy to follow despite of the complicated methods and procedures involved. Nevertheless, I have quite a few questions, both on the technical aspects and on the interpretation.

Thank you for reading our manuscript. The points you raised reveal some important oversights on our part in preparing the manuscript. We hope that the updated text is a more complete summary of the way we conducted our analyses. Thank you for helping us improve our manuscript.

Major Points:

1. One fundamental assumption in this study is that, by constructing task specific regressors and additional intrinsic motivational regressors linked to satiation, and regressing out these effects related to task structure and task events, the residuals contain the fluctuations in task engagement that are intrinsic during the experimental sessions. The reviewer would imagine motion-related artifacts could be a major confound in the residues. It is well established that body motion outside the FOV induces susceptibility changes in the brain, and thus signal changes, even with headpost implantation on the animals’ head that presumably have minimized physical head displacement. There could be plenty of signal fluctuations related to body motion during task execution, which could, of course, generate auto-correlation in the time course, at least in the awake monkey data that reviewer previously looked at. Obviously, the motion is specific to experimental setup and task designs. Persistent autocorrelation in the residuals was used as the major evidence to support the existence of “current engagement”. Somewhat to my surprise, the authors did not mention motion issue in the data analysis. How were potential confounding factors from motion handled?

We appreciate the reviewer’s question and have extensively thought about this topic as it was also one of our major concerns going into the study. We have four major reasons to believe that our findings are not related to motion artifacts.

- 1) Motion artifacts are relatively sudden and fast, corrupting the signal at the time that they happen. However, the disengagement events we identified are associated with robust changes in neural activity that occur prior to the disengagement event itself. By this logic, motion confounds would appear as the absence of the usual motion artefact associated with hand responses, and not as the activity we find. Moreover, we can also see similar activity in periods when we estimated disengagement was likely to happen even if it did not actually happen. Both observations make it unlikely that disengagement-related neural signals are due to motion artifacts. Additionally, we can look at trials

without disengagement, i.e. the periods when disengagements are likely but did not happen, and see the same neural pattern, showing that it cannot all be simply about the motion driven impact of disengagements.

- 2) We have carefully looked at the spatial pattern of neural activity identified by our disengagement regressors and can see no evidence that it “looks” like motion artifacts. Motion artifacts are generally characterized by ring like activity patterns or other scanning protocol related-stripe shapes that do not respect anatomical boundaries. However, our activity patterns are frequently defined by such boundaries.
- 3) We have done extensive preprocessing and taken other measures to remove fMRI data compromised by motion related corruption and signal loss. All our fMRI datasets are pre-processed using the same dedicated pipeline. This pipeline, which is part of the MrCat toolbox (<https://github.com/neuroecology/MrCat>), was described in the original publications for the four tasks we used but we should have also reproduced it in this manuscript. Using this toolbox, we estimate and include 13 motion-related regressors in each (neural) GLM. These regressors are principal components that capture volume-by-volume magnetic field distortions (as induced by limb and body movements) estimated by the pre-processing pipeline.
- 4) We further account for motion by fully excluding low-quality EPI volumes (i.e. volumes suffering from strong artefacts) as identified by the pipeline.

We have updated the manuscript to now include these points. The updated text in the Methods section now reads:

“Moreover, we also included confound regressors to index head motion and volumes with excessive noise. **Motion-related artefacts were captured by including 13 principal components accounting for volume-by-volume magnetic field distortions due to limb and body movements during task performance. Volumes with excessive noise were entirely excluded from the fMRI analysis by including regressors for each flagged volume. Both the 13 principal components and the low-quality volumes were estimated for each session using the MrCat toolbox (<https://github.com/neuroecology/MrCat>) as also described in the original publications for each dataset (Bongioanni et al., 2021; Grohn et al., 2020; Jahn et al., 2022; Khalighinejad et al., 2020, 2022).**”

We now also reference this updated Methods section in the first paragraph of our fMRI Results section:

“As in the behavioral analyses, we constructed a separate neural regression model for each task that captured all aspects of the extrinsic task variables (see *Methods* for the specific models). In addition to these task-specific models, we also included regressors that captured the task engagement factors that we identified in our behavioral analysis (Fig.1C), **and regressors accounting for body and limb motion during task-performance and low-quality volumes (see *Methods* for details).**”

2. The interpretation of the pgACC TUS data. Unlike stimulating sensory or motor systems, stimulating the cingulate cortex could elicit autonomic reaction and complex subjective responses. Such effects are difficult to assess in animals, but have been

reported in humans. In this regard, I would like to point out a paper by Parvizi and Greicius et al. 2013 (The Will to Persevere Induced by Electrical Stimulation of the Human Cingulate Gyrus). Thus, it would appear questionable to attribute the effects observed after acute TUS as simply a change in engagement.

Apologies if our manuscript implied that we believe TUS to simply move a “engagement lever”. We fully appreciate that the impacts of offline TUS for the approximately hour after stimulation are complex and multifaceted. We also agree that the stimulated region has multiple functions and connections that could impact behaviour directly and indirectly. We fully agree with the reviewer that a lot of work needs to be done to understand the reasons why stimulation impacted the observed behaviour (i.e. engagement levels) and subjective changes in the animals may have accompanied the cognitive state they experienced (such as emotion, energy level or ability to focus on task instructions, response cues or other task features). In fact, in future work we will tease out the single neuron level responses and how they change with different levels of task engagement as predicted in our model throughout different medial prefrontal regions to get closer to an answer to the reviewer’s question. However, for the purposes of this study, we tried to stay close to the behaviour that we have measured directly: behavioural engagement.

Following the reviewer’s advice, we are now stating the multiple reasons why stimulation might have impacted engagement in the manuscript. We are now writing in the Discussion:

However, while the stimulated pgACC appeared to track our engagement variable, we acknowledge that several factors, such as emotion, energy level and ability to focus, and subjective emotional responses (Parvizi et al., 2013) may be directly or indirectly linked to engagement. Future research will need to tease apart.

3. More generally, are “engagement” and “disengagement” fundamental constructs of animal behavior or a behavioral expression of some more fundamental constructs? In this regard, is it possible to fit the observed finding in a framework of default mode network and attention network?

We believe that the reasons an animal might disengage with a specific task or behaviour are diverse and not captured by one single mechanism. Here we focused on disengagement in contexts in which the animal was, otherwise, primarily engaged in one experimental task to which they would subsequently return. In the natural world, after disengaging with one task, animals would have likely been able to engage with other longer term behaviours meaningfully during that period of disengagement. Thus, other brain circuits might have played a key role in enabling the animal to not just switch away from the task in hand but to switch to another task. However, this still means that being able to stay on task or becoming disengaged and ready to switching away is likely to be a fundamental process. Its relationship to concepts such as attention and DM networks is an interesting question, we don’t know the answer to as we lack a detailed cognitive understanding of the processes occurring within a DM network, but one could certainly imagine that task negative activity like DM might

meaningfully interact with our ability to continue to stay on task through their opposite impact on fatigue.

We added the following text to the Discussion:

While we could show task-general and robust neural and behavioural patterns related to task engagement, we do not know what cognitive/emotional or otherwise internal construct is driving motivation states, as we cannot ask the animal about their subjective experience. It is possible that other fundamental constructs are linked to our pgACC activity in particular, which in turn relate to the motivational state the animal is in, instead of pgACC driving motivation directly. However, whatever such a fundamental construct might be responsible, it appears intimately linked to motivation across tasks.

Specific points:

1) Figure 1 shows CE, GE and general engagement kernel, do the black dots in Fig. 1D indicate moments of disengagement? It would be helpful to also illustrate the temporal relationship between the negative peaks in CE as shown in Fig. 1D and moments when cues/tasks were presented and executed. Also there is no time scales in the trials (x-axis).

To clarify, yes, the black dots are moments of disengagement. We now describe the relationship between them and CE more thoroughly in the manuscript:

While CE and the disengage choices are closely (inversely) related, CE values are impacted by the degree of predictability of a specific disengagement choice (black dots in Fig.1D vs orange line in Fig.1D).

Regarding the depiction of other events, this is not possible in the plot as it doesn't depict time but is plotted trial by trial. In other words, there is always a cue and a response, but the figure only plots disengagements (as dots) and model predictions.

2) Kernel derivation: from the method description, it is difficult to understand the criteria (if any) to determine alpha in the kernel function. It is task specific, is it also run-specific? How was the data from multiple runs combined?

We concatenate, which means we do not use different kernels for different runs or different animals. Instead, we use one kernel per task. This is done because we want to estimate general engagement, which does not necessarily need to regress out the most variance of engagements on a session-by-session basis but rather should serve as a regressor for the fMRI analysis that captures the general level of engagement on a trial. As such, it should not be 'overfit' on any single session. We apologize for not making this clearer in the manuscript. The updated text in the Methods section now reads:

To compute the task engagement state, we fitted an exponential kernel to our measure of intrinsic task engagement. Specifically, we found the free parameter α that minimised the squared distance between the function $\alpha(1 - \alpha)^{|d|}N$ and the data, where for each trial, d indexes all past and future trials of a session, leaving out the current trial, i.e. $d = \text{first trial}, \dots, -2, -1, 1, 2, \dots, \text{last trial}$, and N is a normalization factor that makes the weights sum up to 1, i.e. $N = \sum(\alpha(1 - \alpha)^{|d|})^{-1}$. We compute α separately for each of our 4 tasks **by finding the α that minimizes this error across all sessions associated with that task. Thus, we overall fit four values of α .**

3) In the data shown in Fig. 2, there are many spotty clusters in the activation maps, which lead to classical questions about false positive/negative and thresholding in fMRI. Given the small number of monkeys involved in each task, a table should be added to summarize number of trials, runs, run duration, number of disengagement events as indicated in the black dots of Fig. 1D. Data like these could indicate the robustness of the statistics.

First, we would like to clarify that by using data across 4 different tasks, our study already has more power than most monkey fMRI studies, and because of this we were able to run whole-brain analyses with 3 levels of hierarchy. Second, spotty activation maps are not necessarily a feature of unreliable data but could instead suggest anatomical idiosyncrasies that might only appear with enough power. Regarding the request for a table, we would like to point the reviewer to Supplementary Figure S1, where we attempted to visualise the information, the reviewer asked for in the form of histograms. These histograms show the proportion of engaged and disengaged trials, their overall count, as well as the length (in log RT) of these engaged and disengaged trials. We have reproduced the figure below. As a lot of what the reviewer requested is already shown in this figure, we added the missing information to the figure legend (also shown below) instead of in a separate table.

For each monkey, we binarized trials into engaged (green) and disengaged trials (red). Trials were coded as disengaged if the monkey responded slower than 3s ($\log(3000\text{ms}) \approx 8$) or did not respond at all (missed trials). Animals that had disengaged on less than 5% of trials were excluded from the study (grey background) **(A)** Shows the data from Jahn et al.. In this experiment a trial timed out if the animal did not respond within 10s. The percentage of ‘disengaged trials’ in the panel title also contain these ‘missed trials’. For monkey 1 we had data from 42 sessions, with an average session duration of 83.22 min (std 13.84 min), for monkey 2 40 sessions (average length 64.04 min; std 15.67 min), and for monkey 3 38 sessions (average length 94.22 min; std 12.16 min) **(B)** shows the data from Bongioanni et al.. Here a trial times out if the animal did not respond within 60s. 2 of the 4 animals were excluded from the study as they did not have enough disengagement trials to qualify. For the 2 included animals, we had 12 sessions for monkey 1 (average length 93.08 min; std 14.99 min) and 12 sessions for monkey 2 (average length 81.77 min; std 14.10). **(C)** shows the data from Khalighinejad et al.. In this task, unlike the other 3 tasks, animals had an incentive to delay their response and respond later in a trial to potentially obtain more reward. As such, we did not count trials with RT > 3s as disengaged trials for this task, and only used the missed trials in which the animals did not respond at all to code disengaged trials. 1 animal had to be excluded because it did not have enough overall disengaged trials. For the 3 included animals, we had 23 sessions for monkey 1 (average length 46.13 min; std 4.63 min), 23 sessions for monkey 2 (average length 48.10 min; std (3.76 min), and 21 sessions for monkey 3 (average length 43.46; std 3.37 min). **(D)** shows the data from Grohn et al.. In this task trials never timed-out if the animal did not respond, but instead the stimuli stayed on the screen until the animal re-engaged with the task. As such, there are no missed trials for this task. 3 of the 6 animals had to be excluded because the total number of disengaged trials was less than 5%. For the 3 included animals, we had 13 sessions for monkey 1 (average length 18.90 min, std 6.10 min), 11 sessions for monkey 2 (average length 23.23 min; std 7.38 min) and 12 sessions for monkey 3 (average length 38.97 min; std 14.62 min).

4) TUS effects on disengagement. Of the 4 monkeys received TUS (Fig. S5), only monkey 1 appears substantially deviate from sham stimulation; monkey 3 is basically non-differentiable. This beg the questions of TUS data interpretation.

While it is true that one monkey is showing an impressive impact of TUS throughout the experiment, we would like to point out that all 4 monkeys have the smallest duration of motivational collapses in the first half of the experiment after pgACC TUS.

Statistically, while we agree that the data for monkey 3 subjectively does not look like there is a strong effect, the statistical test we are conducting is run on the group level across all 4 animals. Our linear model includes random effects for the intercept (which effectively baselines the level of engagement of each animal), and stimulation side (which allows for the effect to vary across animals around the group level effect). As such, finding an overall significant effect on the group level, as we do, constitutes enough evidence across all four animals to reject the null hypothesis that stimulating pgACC has the same effect on engagement as the control regions, while still allowing the effects to vary at the subject level.

5) Trial vigor: response times were used to index trial vigor. Results from this analysis was used as a control to support the main claim. However, by careful examination of Fig. S4, cerebellum seems heavily involved in "trial vigor", is there any explanation for this finding?

We did not focus on the cerebellum but yes, there is an extensive literature on motor control, response speed and practice, linked to the Cerebellum, which could explain its activity in high vigour trials. The fact that vigour and engagement are clearly dissociable with vigour being in more "motoric" region further supports our claim of distinct neural processes being in play. We were only able to find activity related to trial vigour in the Cerebellum, but no activity related to state vigour. Moreover, trial vigour activity could only be found when time-locking to decision-prompt (see Figures S4 and S5). As such, it is likely that the activity in the Cerebellum is related to executing an action rather than a general tendency of being vigorous (which would entail sustained activity that is also indexed by state vigour).

6) A brief explanation of the 4 experimental designs should be warranted in the supplemental such that the potential readers do not have to go back previous publications to look for details.

We have added a supplementary text that contains the experimental descriptions of the four tasks we used. We now also reference this supplementary text throughout the main text.

Reviewer #3 (Remarks to the Author):

The manuscript addresses brain activity associated with (dis)engagement from task in four fMRI studies carried out in macaque monkeys. Disengagement was defined as trials with no response or (in three tasks) trials with very long reaction times. Task-related activity and other factors likely reflecting fatigue, satiety, task history, RT variability etc. were regressed out. Maintenance of task performance was associated with activity in the pregenual cingulate cortex. Midcingulate activity was associated with impending task disengagement. Transcranial ultrasound stimulation of the pgACC reduced the occurrence of task disengagement in the first 20 min after sonication in an offline stimulation protocol. This suggests that pgACC is causally related to task engagement. Importantly sonication of control regions, particularly the basal forebrain, did not have reliable effects on the engagement/disengagement patterns.

The authors address an interesting and timely topic that is difficult to study in humans. The methodological approach is innovative and convincing, given that the results are based on four different tasks. The results may open a new path of research, which should be related to other research topics such as fatigue, lapses of attention, mental effort and mind wandering.

I have a few minor issues and suggestions that may be helpful in a revision of the manuscript.

1. The abstract suggests: "We identified consistent patterns of behavior predictive of impending task disengagement". I was wondering which behavioral patterns the authors relate to. The autocorrelation analysis suggests that engagement and disengagement phases are clustered. But are there any other parameters that might predict an impending cluster of disengagement or sudden disengagements (e.g. RT slowing, increased RT variance, errors)?

Apologies for the bad phrasing. We were indeed referring to our autoregressive model here. Task features did indeed predict lapses - as one would expect a non-rewarded monkey sometimes gives up out of frustration. We are not specifically teasing apart all the task model related disengagement as the tasks varied in their design too much to be directly comparable. However, our autoregressive model did predict additional lapses through the fact that they were clustered. Regarding sudden disengagements i.e. engagement shifts (or engagement lapses), those were, by our definition, completely unpredicted and interestingly led to quite specific neural activity. We have changed the sentence in the abstract to **"We identified consistent autocorrelation in task disengagement."**, which we hope better reflects what we intended to say.

2. I found the direction of the fMRI and TUS findings in pgACC somewhat intriguing. While fMRI data suggest an association of increased pgACC activity with task engagement (and a number of neuroimaging findings consistent with this are cited in the ms) TUS seems to reduce task disengagements. As stated in the MS it seems unlikely that ultrasound effects mimic physiological activity such that I would have expected a detrimental effect of TUS in pgACC on task engagement. Some more elaboration on these effects might help readers less familiar with the methods interpreting the findings.

This is indeed an intriguing question. However, one should not expect TUS to cause a "virtual lesion". As such, we note two important points here. 1) We still don't know how TUS works physiologically and it is possible that it works very differently from direct neurostimulation (TMS or electrode stimulation) and more as a neuromodulator. That is, TUS could have made the pgACC more receptive to its inputs, boosting its impact on the rest of the brain as TUS was done offline prior to the task. This interpretation is in line with recordings made in the original dataset we used by Khalighinejad et al.: In their paper, they show that TUS can increase the connectivity between the stimulated area and other areas (Fig 7B). 2) We don't know what relationship there might be between a general increase or decrease in activity and

motivation itself. It could be that offline TUS depresses “task irrelevant” neuronal “noise”, making the animal more focused at the task at hand for a while. We also offer some pathway specific explanations in the discussion, but those are still speculative at this stage.

Relatively few behavioral experiments have focused on the macaque pgACC and previous behavioral analysis approaches have not allowed identification of clear changes in task engagement (Khalighinejad et al., 2020) of the sort that we were able to identify here. However, it has been reported that electrical microstimulation of the macaque pgACC during a cost/benefit decision making task led to fewer decisions to pay higher costs (enduring air puffs) to obtain higher rewards (more juice) (Amemori & Graybiel, 2012). If pgACC is not only responsible for setting the general willingness to endure costs for benefits during choices but also responsible for setting the general level of engagement, then our results and these previous findings can be reconciled. However, it is important to note that TUS is unlikely to recreate patterned excitation of specific neurons that can be induced by microstimulation but rather it may be more likely to disrupt the endogenous activity patterns within a brain region (Folloni et al., 2021; Verhagen et al., 2019). In the rat, optogenetic inhibition of the projections from the homologue of pgACC (Vogt, 2009) – often called the prelimbic cortex – to the striosome compartment of the striatum similarly leads rats to be more likely to pay the cost of engaging in a trial in order to obtain a reward (Friedman et al., 2015). This occurs because pgACC outputs synapse with inhibitory interneurons in the striosome which, in turn, connect with striatal projection neurons. Thus, disrupting pgACC leads to the release of striatal projection neurons from inhibition. As noted, striosomal projection neurons are distinguished by their unique anatomical connections to regions such as the dopaminergic midbrain. In summary, pgACC TUS or pgACC optogenetically mediated inhibition in monkeys and rats respectively make animals more likely to engage in an effortful task to obtain reward or to take a costly action to obtain reward. Both interventions may resemble one another in leading to the release of striatal projection neurons from inhibition and, as a consequence, changes in dopamine levels.

Beyond our speculations only future research can completely clarify this question by doing post TUS fMRI recordings and see the neural impact of TUS, while physiological studies should further tease apart the biophysical impact of different TUS protocols themselves.

We now discuss this further in the MS:

While we found an impact of TUS on behaviour, we can only speculate about the physiological mechanism(s) at this stage. Hopefully future research on the physiological impact of TUS and post stimulation fMRI will be able to shed some light onto this question, in particularly why TUS can lead to apparent enhancement of behaviours associated with the stimulated brain regions, despite TUS not mimicking neural activity.

3. A number of previous papers might be relevant for this study and the authors might consider discussing them:

- Magno E, Foxe JJ, Molholm S, Robertson IH, Garavan H (2006) The anterior cingulate and error avoidance. *J Neurosci* 26:4769-4773. This study might support the finding that midcingulate activity might be related to sudden shifts in engagement. Here, midcingulate and anterior insular activity was related to "escaping" trials that were expected to be particularly effortful and perhaps less likely to be successful.

- Eichele T, Debener S, Calhoun VD, Specht K, Engel AK, Hugdahl K, von Cramon DY, Ullsperger M (2008) Prediction of human errors by maladaptive changes in event-related brain networks. *Proc Natl Acad Sci U S A* 105:6173-6178. Here slow and systematic changes in brain activity preceding erroneous responses are shown (increasing activity not in pgACC but in posterior cingulate/precuneus and decreasing task-related lateral and medial frontal activity) which was interpreted as accompanying gradual disengagement from task. There are also a number of EEG studies showing systematic changes of brain activity predicting errors, but they are less informative regarding the underlying functional neuroanatomy.

- Weissman DH, Roberts KC, Visscher KM, Woldorff MG (2006) The neural bases of momentary lapses in attention. *Nat Neurosci* 9:971-978. and Chee MW, Tan JC, Zheng H, Parimal S, Weissman DH, Zagorodnov V, Dinges DF (2008) Lapsing during sleep deprivation is associated with distributed changes in brain activation. *J Neurosci* 28:5519-5528.

The definition of lapses of attention is quite similar to the RT-based definition of disengagement used in the present manuscript.

We apologise for the omissions for this relevant literature. The research of errors and error monitoring is definitely relevant for our study and so is their localization within the MCC. We would like to point out, however, that during our analyses we attempt to regress out activity associated with reward, and therefore negative feedback.

We now cite the references the reviewer suggested in the manuscript. In particular, we added the following text to the discussion:

Our results have obvious links to a large body of work on error monitoring (Magno et al., 2006) and performance lapses (Eichele et al., 2008) in humans that have identified ACC/MCC and pre-SMA as relevant regions for both error monitoring as well as post error adaptation effects. While our lapses are a complete disengagement from the task, not an error *per se*, the overlap between the anatomical location of the effects reported here and the effects previously reported is intriguing, suggesting common mechanisms may prevent disengagement, maintain engagement with the current task, and mediate performance monitoring for errors and post-error adaptation and return to task performance.

4. While the authors suggest that the vigor-related analysis does not show activity overlapping with the (dis)engagement-related activity, I am not fully convinced when

comparing the figures. What is called frontopolar cortex in Fig S3A seems to at least partly overlap with the activity around the pgACC cluster in the main analysis.

For clarity we show above the relevant images in question here. First, on the left, is the disengagement effect, which was an effect that was linked to a pattern of behaviour sustained across not just the current trial but also over previous and subsequent trials. This corresponds to figure 2C in the manuscript. Next, at the centre, we show the effect of vigour, which was considerably weaker and less spatially extensive. We think that this panel, which corresponds to Figure S4A, is the one that the reviewer is referring to. It illustrates the effect of vigour before the current trial (i.e. in the ITI preceding it), but no effect of vigour can be found in this region as the animals are responding. It also is not centred in the pgACC but on a part of the frontopolar cortex. Even if there is some small amount of overlap between the edges of the pgACC activation related to engagement and the frontopolar cortex activation for vigor, it is very limited in degree. We apologise if we did not acknowledge this limited overlap when we were focussed on pointing out that there are important differences not just in behavioural indices of vigour and disengagement but also in the neural activity linked to vigour and disengagement. The vigour-related result in the panel on the right above (which corresponds to Figure S4D) is the result that is anatomically most similar to our pgACC engagement effect in the panel on the left; it reflects the degree to which the vigor is improving (future responses will be more vigorous than past responses), in other words, it is a bit like the first derivative of the vigor, but as the effect is small, we did not highlight this further in the manuscript. We now changed the MS text in the following way:

While vigor and engagement were associated with different behavioral indices and had correspondingly distinguishable relationships with brain activity, some of the effects were adjacent in the brain. However, vigor effects often appeared to be mostly linked with the vigor level on the preceding the current trial (see Fig S4 and S5) or with increases of vigor that were about to occur (Fig S4D future vigor – past vigor). By contrast, engagement effects reflected stable patterns of behavior sustained over several trials.

5. What could be the reason for the much stronger *t*-statistics in the 4th study (Fig 3, Bongioanni et al.)?

We unfortunately do not have enough data to conduct any proper analyses, but we would like to point out that the behavioural autocorrelation we examine is the strongest

for that task (Fig 1C). This autocorrelation is then used to construct the 'general engagement' contrast we use in the whole-brain analysis, and it is for this 'general engagement' contrast that we observe the strong effect for Bongioanni et al. As such, it seems plausible that the strong effect is simply due to our model working best for that task. Additionally, as can be seen in Figure S1, the proportion of disengaged trials is larger for the two monkeys that participated in Bongioanni et al. (around 20%) than in other tasks (where it is around 10%). This might also lead to the neural signals appearing stronger in these two animals.

6. While regressing out fatigue effects does make sense in the present study, the topics fatigue, task engagement, and apathy are related to each other. I would appreciate a discussion on the interaction between fatigue and task engagement. Perhaps the results by M. Apps might be of relevance as well.

We fully agree with the reviewer and are now discussing the relationship of our findings with fatigue models more thoroughly. Specifically, we believe those findings to be complementary, offering interesting models of how task related activity can predict disengagement (e.g. our regressors of trial number, rewards received etc), while not being sufficient to explain ALL disengagements (as those require capturing further elements such as our intrinsic fluctuation models). One exciting future prospect would be to combine both approaches and see how our intrinsic model modulates the degree to which fatigue accumulates and other task features are processed.

We now discuss this in the MS:

While we focused on the task independent elements of motivation and engagement there is, of course, a large body of work on motivation, fatigue and apathy based on effort and cost models (Blain et al., 2019; Matthews et al., 2023; Müller et al., 2021; Müller & Apps, 2019; Wiehler et al., 2022). Future research could potentially combine both approaches (intrinsic/task independent and task driven motivational fluctuations) to get a more comprehensive picture of their interplay.

7. The fMRI setup precludes any investigation on what the monkeys might do when disengaging from task. Are there any systematic reports in the literature what macaques would engage in when interrupting task-engagement? Could disengagement from one task also be driven by the impression that another task has become more important at this point? This would be substantially different from a fatigue-driven disengagement or from mind wandering.

This is indeed an interesting question, but sadly we did not have systematic video recording and video classification to answer this question conclusively. Anecdotally, we note that we observed that they either shut their eyes, sleep, looked around a bit, scratched themselves, or simply did very little until the task caught their attention again.

REVIEWERS' COMMENTS

Reviewer #1 (Remarks to the Author):

The researchers explored the inherent aspect of engagement in animals by observing the behavior of macaques in four distinct tasks while simultaneously recording fMRI signals. The updated manuscript effectively addressed my initial concern.

Reviewer #2 (Remarks to the Author):

The authors did a good job in addressing my questions. I have no further comments.

Reviewer #3 (Remarks to the Author):

The revised version and the authors' responses to my comments have convinced me. I think, this is a really innovative approach to analyzing task (dis)engagement. In my view, the manuscript is ready for publication.